# FXR1 regulates transcription and is required for growth of human cancer cells with *TP53/FXR2* homozygous deletion

Yichao Fan[1], Jiao Yue[1], Mengtao Xiao[1], Han Han-Zhang[1], Yao Vickie Wang[1], Chun Ma[1], Zhilin Deng[1], Yingxiang Li[2], Yanyan Yu[1], Xinghao Wang[1], Shen Niu[1], Youjia Hua[1], Zhiping Weng[2], Peter Atadja[1], En Li[1]*, Bin Xiang[1]*

[1]Epigenetic Discovery, China Novartis Institutes for BioMedical Research, Shanghai, China; [2]Department of Bioinformatics, Tongji University, Shanghai, China

**Abstract** Tumor suppressor p53 prevents cell transformation by inducing apoptosis and other responses. Homozygous *TP53* deletion occurs in various types of human cancers for which no therapeutic strategies have yet been reported. TCGA database analysis shows that the *TP53* homozygous deletion locus mostly exhibits co-deletion of the neighboring gene *FXR2*, which belongs to the Fragile X gene family. Here, we demonstrate that inhibition of the remaining family member FXR1 selectively blocks cell proliferation in human cancer cells containing homozygous deletion of both *TP53* and *FXR2* in a collateral lethality manner. Mechanistically, in addition to its RNA-binding function, FXR1 recruits transcription factor STAT1 or STAT3 to gene promoters at the chromatin interface and regulates transcription thus, at least partially, mediating cell proliferation. Our study anticipates that inhibition of FXR1 is a potential therapeutic approach to targeting human cancers harboring *TP53* homozygous deletion.

DOI: https://doi.org/10.7554/eLife.26129.001

*For correspondence: en.li@ novartis.com (EL); bin.xiang@ novartis.com (BX)

## Introduction

p53, a critical tumor suppressor, primarily serves as a transcription factor to control various cellular stress-response signaling pathways, including cell cycle arrest, DNA repair, senescence and apoptosis (*Vousden and Lane, 2007*). Loss of p53 function through mutation or deletion of its encoding *TP53* is a common feature in a majority of human cancers, resulting in the escape from tumor-suppressor activities. Numerous strategies have been explored to reverse dysregulated p53 suppressor function, including stabilizing p53 expression by antagonizing the p53–MDM2 interaction in cancers harboring normal *TP53* copy number, and restoring p53's tumor suppressor activity in *TP53*-mutated cancer (*Khoo et al., 2014*; *Soragni et al., 2016*). In cancer, tumor suppressor genes are often inactivated by genomic deletions that encompass the deletion of neighboring genes. Such bystander genes often belong to multigene families; therefore, their deletion is tolerated due to genetic redundancy. The newly proposed 'collateral lethality' concept suggests that such passenger deletion predisposes cancer cells to vulnerabilities that are induced by further inhibition of the remaining genes in the family, whose functions are essential and redundant (*Muller et al., 2012*, *2015*; *Nijhawan et al., 2012*). This concept therefore opens the avenue to anti-cancer drug development targeting cancers containing co-deletion of tumor suppressor genes and neighboring genes without affecting wild-type cells. Drugging *TP53*-deleted cancers had been a challenging task until a recent report demonstrated inhibition of a neighboring gene *POLR2A*, which is located about 200 kb downstream of *TP53* on chromosome 17 and undergoes heterozygous deletion in colorectal cancers containing *TP53* heterozygous deletion (*Liu et al., 2015*). Homozygous deletion, resulting in inactivation of both alleles, occurs less frequently and is more focal than heterozygous deletion.

**eLife digest** Healthy human cells employ many tricks to avoid becoming cancerous. For example, they produce proteins known as tumor suppressors, which sense if a cell shows early signs of cancer and instruct the cell to die. A gene known as *TP53* produces one of the most important tumor suppressor proteins, and this gene is inactive or missing in many types of human cancer.

Treating cancers that have completely lost the *TP53* gene is particularly difficult. One way to develop new treatments for these conditions would be to target other proteins that these cancers need to survive; but these proteins first need to be identified.

Fan et al. have now identified one such protein in human cancer cells lacking *TP53*. Searching databases of DNA sequences from human cancer cells revealed that those without the *TP53* gene often also lose a neighboring gene called *FXR2*. Cancer cells survive without *FXR2* because a similar gene, called *FXR1*, can compensate. Fan et al. therefore decided to experimentally lower the activity of the *FXR1* gene and, as expected, cancer cells without *TP53 and FXR2* stopped growing. Normal cells, on the other hand, were unaffected by the deletion of the *FXR1* gene since *FXR2* is still there. This phenomenon, in which cancer cells become vulnerable after the loss of certain genes but only because they have already lost important tumor suppressors, is called "collateral lethality". Further experiments showed that the protein encoded by *FXR1* coordinates with other proteins to activate genes that contribute to cell growth.

These findings suggest new ways to treat human cancers that have lost *TP53*. For example, if scientists can find small molecules that inhibit the protein encoded by *FXR1* and show that these molecules can block the growth of tumors lacking *TP53* and *FXR2*, this could eventually lead to a new anticancer drug. However, like any new drug, these small molecule inhibitors would also need to be extensively tested before they could be taken into human clinical trials.

DOI: https://doi.org/10.7554/eLife.26129.002

There is no documented therapeutic strategy targeting homozygous *TP53*-deleted cancers. *POLR2A* is co-deleted in a majority of tumors with *TP53* homozygous deletion, and thus its inhibition would not be relevant. *FXR2* (Fragile X-related Protein 2, also known as FXR2P), located 100 kb downstream of *TP53*, is also a neighboring gene of *TP53* at chromosome 17p13.1. It belongs to the fragile X gene family that has essential functions in binding and regulating mRNA stability, transportation and translation (*Ascano et al., 2012*; *Chen et al., 2014*; *Darnell et al., 2001*; *Siomi et al., 1996*). In this study, we investigated whether *FXR2* passenger deletion at the *TP53* homozygous deletion locus would result in subsequent cancer-specific vulnerability to inhibition of its family member, FXR1 (Fragile X-related Protein 1, also known as FXR1P).

The fragile X gene family contains three mammalian members, including fragile X mental retardation protein FMR1 (also called as FMRP) and its structural homologs FXR1 and FXR2 (*Siomi et al., 1993*, *1995*). These proteins are highly conserved in many species and share a high degree of sequence similarity in major functional domains, including tandem Tudor, KH and RGG box domains (*Kirkpatrick et al., 2001*). They all participate in RNA-binding, regulation of mRNA metabolism, ribosome-binding, and translation (*Ascano et al., 2012*; *Chen et al., 2014*; *Darnell et al., 2001*; *Siomi et al., 1996*). These proteins contain nuclear localization and nuclear export signals which allow them to be shuttled between cytoplasm and nucleus (*Eberhart et al., 1996*). FMR1, which is highly expressed in neurons, plays a critical role in synaptic plasticity and its silencing results in Fragile X Syndrome, an inherited intellectual disability and the major cause of autism (*Consortium, 1994*; *Darnell et al., 2011*; *Santoro et al., 2012*; *Verkerk et al., 1991*). FXR1 is ubiquitously expressed and has potential roles in cardiac and skeletal muscle development (*Huot et al., 2005*; *Mientjes et al., 2004*; *Van't Padje et al., 2009*; *Whitman et al., 2011*). Increasing evidence suggests that FXR1 and FXR2 possess both common and distinct functions in post-transcriptional regulation (*Ascano et al., 2012*; *Cavallaro et al., 2008*; *Darnell et al., 2009*; *Say et al., 2010*; *Xu et al., 2011*). FXR1 copy number was amplified and demonstrated oncogenic activity in lung squamous cell carcinoma (*Comtesse et al., 2007*; *Qian et al., 2015*). A recent study suggested that FXR1 downregulates p21 by binding and reducing its mRNA stability and/or by modulating p53 expression to avoid senescence in cancer (*Majumder et al., 2016*). At the

N-terminus, all three members of the FMR1 family contain a tandem Tudor domain belonging to the Royal family of chromatin-binding proteins (*Adams-Cioaba et al., 2010*; *Hu et al., 2015*; *Myrick et al., 2015*). The tandem Tudor domain in FMR1 was reported to recognize methylated lysine on histones at the chromatin interface, thus resulting in the protein's subsequent localization to the DNA damage loci, thereby facilitating the repair process (*Adams-Cioaba et al., 2010*; *Alpatov et al., 2014*). Nevertheless, the function of FXR1 in cancer and the underlying molecular mechanisms remain elusive.

From an analysis of The Cancer Genome Atlas (TCGA) and Cancer Cell Line Encyclopedia (CCLE) databases, we observed that *FXR2* undergoes concomitant homozygous deletion in combination with *TP53* homozygous deletion in a significant proportion of human cancers. Our data show that FXR1 inhibition blocks cell growth in *TP53* and *FXR2* co-deletion cell lines, but not in copy number normal or single deletion cell lines. The results were further confirmed in CRISPR-Cas9-generated *TP53* and *FXR2* knockout cell clones. Through a comprehensive analysis of chromatin immunoprecipitation followed by mass spectrometry (ChIP-MS) and ChIP coupling with high-throughput sequencing (ChIP-seq), we uncovered the molecular mechanism through which FXR1 regulates cell proliferation. FXR1 is located in the gene promoter region together with histone H3 lysine 4 trimethylation (H3K4me3), and recruits transcription factor STAT1 or STAT3 at gene promoters to facilitate transcription regulation. In an agreement with a role in the FXR1-associated mechanism, inhibition of STAT1 or STAT3 target genes can also reduce cell proliferation. Taken together, our findings not only demonstrate the possibility of inhibiting FXR1 in *TP53* and *FXR2* homozygous co-deletion cancers as a potential treatment option, but also reveal the novel role of FXR1 in gene transcription.

## Results

### *FXR1* knockdown inhibits cell proliferation in *TP53/FXR2* co-deletion cancers

Copy number data in various human tumors published in TCGA database (access through cBioPortal (http://www.cbioportal.org; accessed 18 Aug 2016) (*Gao et al., 2013*) show that homozygous *TP53* genomic deletion occurs in various human cancers, at frequencies ranging from 1% to 15% (*Figure 1A*, *Figure 1—figure supplement 1A*). One of the neighboring genes, *FXR2*, located around 100 kb downstream of *TP53* at chromosome 17p13.1, undergoes concomitant deletion in most tumors carrying *TP53* homozygous deletion with only a few exceptions (*Figure 1A and B*). Consistently, in the CCLE database (*Barretina et al., 2012*), the majority of cancer cell lines carrying *TP53* homozygous deletion also contain *FXR2* deletion (*Figure 1—figure supplement 1B*). However, *FXR2* single deletion is rarely observed in human tumors. In *TP53/FXR2* co-deletion tumors, the copy number of *FXR1* and *FMR1* (*Figure 1A*, *Figure 1—figure supplement 1B*) is largely unaltered. Of note, *FXR1* or *FMR1* copy number gain was also observed in tumors, although this wasmutually exclusive with *TP53/FXR2* deletion (*Figure 1A*, *Figure 1—figure supplement 1B*). The recent discovery of the collateral lethality concept prompted us to hypothesize that concomitant deletion of passenger *FXR2* in *TP53*-deleted cancer cells might make cell growth dependent on FXR1. We therefore tested whether TP53/FXR2 co-deletion renders cancer cells sensitive to FXR1 inhibition. We selected four cancer cell lines harboring co-deletion of *TP53* and *FXR2*: KATOIII, HL-60, H358, and KMS-11, as well as four cell lines harboring the normal copy number of *TP53* and *FXR2*: MKN45, AGS, HepG2, and A549. The mRNA and protein level of p53, FXR2, FXR1 and FMR1 were assessed using q-RT-PCR and Western Blot (WB), respectively. Our data showed that cancer cells with homozygous co-deletion of *TP53* and *FXR2* exhibit either absent or significantly lower mRNA and protein levels of p53 and FXR2. By contrast, the copy-number-normal and co-deleted cell lines have comparable FXR1 and FMR1 levels (*Figure 1—figure supplement 2*). It is worth noting that p53 protein levels were assessed under a stress condition induced by doxorubicin.

Next, we monitored cell proliferation rate upon FXR1 inhibition using inducible short hairpin RNAs (shRNAs). Five doxycycline (Dox)-inducible FXR1 shRNAs were tested and most of them resulted in robust downregulation of FXR1 protein level and exhibited anti-proliferative activity in the *TP53/FXR2* co-deleted cancer cell line KATOIII, but not in the copy-number-normal cell line MKN45 (*Figure 1—figure supplement 3A*). We selected shRNA 2 and 3 (FXR1-sh2, FXR1-sh3),

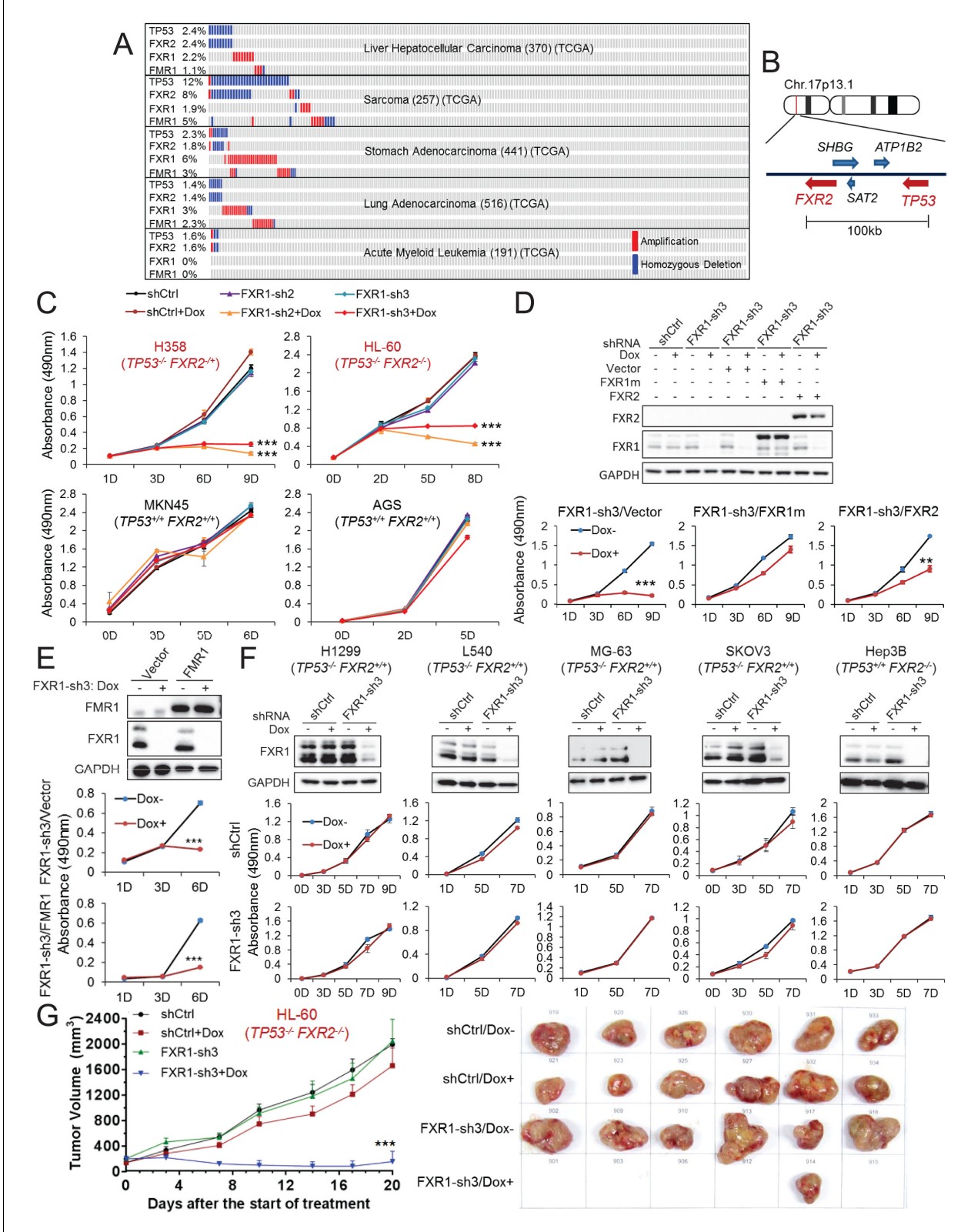

**Figure 1.** FXR1 knockdown inhibits cell proliferation in cancer cells containing both *TP53* homozygous deletion and *FXR2* passenger deletion. (**A**) Copy number alterations in *TP53*, *FXR2*, *FXR1*, and *FMR1* in various types of human tumors based on TCGA data analysis. (**B**) Schematic diagram of *FXR2* and *TP53* location at chromosome 17p13.1. (**C**) Cell proliferation rate upon FXR1 inducible knockdown. Measurements of change in cell proliferation rate induced by control shRNA (shCtrl) or by FXR1 shRNA (FXR1-sh2, FXR1-sh3) upon doxycycline (Dox) treatment (at indicated days) in *TP53* and *FXR2* co-

*Figure 1 continued on next page*

*Figure 1 continued*

deletion-containing cancer cell lines HL-60 and H358 (upper panel) and in the copy-number-normal cancer cell lines MKN45 and AGS (lower panel) using an MTS assay. Data represent the mean ± s.d. of three independent experiments. (D) Rescuing FXR1-sh3-induced anti-proliferation by ectopic expression of FXR1 or FXR2 in *TP53/FXR2* deletion cancer cell line KATOIII. Upper panel: protein levels of a shRNA-resistant form of full-length FXR1 (FXR1m_a) or of FXR2 upon knockdown of endogenous FXR1 in a Western Blot (WB) assay. Lower panel, cell proliferation in vector, FXR1m_a, or FXR2 ectopic expression cells upon Dox-induced FXR1 knockdown in the MTS assay. Data represent the mean ± s.d. of three independent experiments. (E) Rescuing FXR1-sh3-induced anti-proliferation by ectopic expression of FMR1 in the *TP53/FXR2* deletion cancer cell line H358. Upper panel: ectopic expression level of FMR1 and knockdown efficiency of FXR1 by Dox-induced shRNA. Lower panel, cell proliferation in H358 cells ectopically expressing vector or FMR1 upon Dox-induced FXR1 knockdown in the MTS assay. Data represent the mean ± s.d. of three independent experiments. (F) Cell proliferation change upon FXR1-inducible knockdown in cancer cells with a *TP53* single deletion (H1299, L540, MG-63, SKOV3) or an *FXR2* single deletion (Hep3B). Upper panel, FXR1 protein level upon shCtrl or FXR1-sh3 treatment. Lower panel, cell proliferation measurement in MTS assay. Data represent the mean ± s.d. of three independent experiments. (G) Tumor growth in cancer cell xenograft upon FXR1 knockdown. Left, growth curves of xenograft tumors derived from subcutaneously implanted HL-60 cells stably expressing shCtrl or FXR1-sh3 upon Dox treatment. Tumor volume (mm$^3$) represents the mean ± s.e.m of six mice for each group after the indicated number of days with Dox treatment. Right, the tumor sizes on the termination day. The cell proliferation rate was determined by measuring absorbance at 490 nm in the MTS assay (Y axis). *p<0.05, **p<0.01, ***p<0.001. Also see *Figure 1—figure supplements 1, 2, 3, 4, 5* and *6*.

DOI: https://doi.org/10.7554/eLife.26129.003

The following source data and figure supplements are available for figure 1:

**Source data 1.** Source data for *Figure 1*.
DOI: https://doi.org/10.7554/eLife.26129.010

**Figure supplement 1.** Passenger deletion of *FXR2* in *TP53* homozygous deletion cancer.
DOI: https://doi.org/10.7554/eLife.26129.004

**Figure supplement 2.** Expression of p53, FXR2, FXR1 and FMR1 in human cancer cell lines.
DOI: https://doi.org/10.7554/eLife.26129.005

**Figure Supplement 2—source data 1.** Source data for *Figure 1—figure supplement 2*.
DOI: https://doi.org/10.7554/eLife.26129.011

**Figure supplement 3.** FXR1 knockdown inhibits cell proliferation in *TP53* homozygous deletion cancers upon passenger deletion of *FXR2*.
DOI: https://doi.org/10.7554/eLife.26129.006

**Figure supplement 3—source data 1.** Source data for *Figure 1—figure supplement 3*.
DOI: https://doi.org/10.7554/eLife.26129.012

**Figure supplement 4.** FXR1 knockdown inhibits cell growth in a Matrigel-based three-dimensional assay.
DOI: https://doi.org/10.7554/eLife.26129.007

**Figure supplement 5.** Rescuing FXR1 shRNA-induced anti-proliferation by ectopic expression of FXR1.
DOI: https://doi.org/10.7554/eLife.26129.008

**Figure supplement 5—source data 1.** Source data for *Figure 1—figure supplement 5*.
DOI: https://doi.org/10.7554/eLife.26129.013

**Figure supplement 6.** The effect of FMR1 knockdown on cancer cell proliferation.
DOI: https://doi.org/10.7554/eLife.26129.009

**Figure supplement 6—source data 1.** Source data for *Figure 1—figure supplement 6*.
DOI: https://doi.org/10.7554/eLife.26129.014

which showed profound knockdown efficiency for the following studies. Consistently, FXR1 inhibition only suppressed cell proliferation in the cell lines with *TP53/FXR2* co-deletion (H358, HL-60, KATOIII and KMS-11) but not in the copy-number-normal cells (AGS, A549, MKN45 and HepG2), as indicated in *Figure 1C* and also in *Table 1* and *Figure 1—figure supplement 3*. Furthermore, we used cell imaging and Matrigel assays to confirm FXR1-knockdown-induced anti-proliferative effects (*Figure 1—figure supplement 4*). To determine whether the observed cell growth reduction is attributed to FXR1 inhibition, we rescued FXR1 expression by ectopic expression of shRNA-resistant FXR1 (harboring mutations on the shRNA-targeting sequence, FXR1m) (*Figure 1—figure supplement 5*). In multiple cancer cell lines, the growth rate was recovered to a level comparable to that of the control, indicating that FXR1 inhibition was responsible for the growth phenotype (*Figure 1—figure supplement 5*, *Figure 1D*, Figure 3B). Consistent with the FXR1m data, expression of FXR2 also rescued the anti-proliferative phenotype induced by FXR1 inhibition, suggesting that FXR2 and FXR1 have redundant functions (*Figure 1D*). However, ectopic expression of FXR2 only showed partial rescue compared with FXR1m, suggesting that the proteins may also have some distinct functions. Interestingly, FMR1 did not rescue FXR1 knockdown-induced anti-proliferation (*Figure 1E*).

**Table 1.** Summary of FXR1-knockdown-induced anti-proliferation in cancer cell lines and CRISPR-Cas9-engineered cell clones.
The cell line or CRISPR clone name, cancer type, *TP53* and *FXR2* copy number alteration (based on the CCLE database or knockout confirmation), FXR1-sh3 knockdown (KD) efficiency (determined by q-RT-PCR and WB), and FXR1 KD-induced anti-proliferation efficiency are listed. Dark green, anti-proliferation efficiency >60%. Yellow green, anti-proliferation 30–60%. Yellow, anti-proliferation <30%. Data of KD efficiency and anti-proliferation efficiency represent means of at least three independent experiments. Del, deletion; WT, copy number normal; KO, knockout.

| | Cell | Cancer | TP53 | FXR2 | FXR1 KD efficiency (%) | FXR1 KD induced anti-proliferation (%) |
|---|---|---|---|---|---|---|
| Cancer cell line | HL-60 | Acute myeloid leukemia | Del | Del | 77.6 | 73.4 |
| | KATOIII | Gastric | Del | Del | 83.5 | 76.4 |
| | KMS-11 | Multiple myeloma | Del | Del | 95.7 | 88.3 |
| | H358 | Lung | Del | Del | 79.9 | 95.3 |
| | H1299 | Lung | Del | WT | 96.4 | −1.1 |
| | L540 | Hodgkin lymphoma | Del | WT | 94.9 | −5.8 |
| | MG-63 | Osteosarcoma | Del | WT | 96.8 | −4.5 |
| | SKOV3 | Ovarian | Del | WT | 89.0 | −7.3 |
| | Hep3B | Liver | WT | Del | 94.0 | −0.8 |
| | A549 | Lung | WT | WT | 72.7 | 15.5 |
| | AGS | Gastric | WT | WT | 73.6 | 12.8 |
| | HepG2 | Liver | WT | WT | 92.0 | 2.8 |
| | MKN45 | Gastric | WT | WT | 71.5 | 3.4 |
| CRISPR-Cas9 KO clone | DKO1 | Gastric | WT | WT | 88.8 | 1.9 |
| | DKO5 | Gastric | WT | WT | 86.0 | 2.7 |
| | DKO11 | Gastric | KO | KO | 79.0 | 49.8 |
| | DKOL7-1 | Gastric | KO | KO | 92.0 | 68.5 |
| | DKOL7-3 | Gastric | KO | KO | 87.5 | 45.3 |
| | FXR2 KO10 | Gastric | WT | WT | 80.6 | 34.1 |
| | FXR2 KO11 | Gastric | WT | WT | 75.8 | 19.0 |
| | FXR2 KO17 | Gastric | WT | WT | 77.2 | 6.8 |
| | FXR2 KO5 | Gastric | WT | KO | 81.3 | 16.2 |
| | FXR2 KO16 | Gastric | WT | KO | 72.9 | 15.6 |
| | TP53 KO3-3 | Gastric | WT | WT | 51.1 | −5.3 |
| | TP53 KO3-10 | Gastric | WT | WT | 92.9 | −5.8 |
| | TP53 KO3-6-1 | Gastric | KO | WT | 85.3 | 34.0 |
| | TP53 KO3-6-2 | Gastric | KO | WT | 81.4 | −5.3 |
| | TP53 KO3-14-6 | Gastric | KO | WT | 86.2 | 12.5 |

DOI: https://doi.org/10.7554/eLife.26129.015
The following source data available for Table 1:
**Source data 1.** Source data for *Table 1*.
DOI: https://doi.org/10.7554/eLife.26129.016

Furthermore, FMR1 knockdown had no impact on proliferation in both *TP53*/*FXR2* copy-number-normal and co-deleted cancer cells (*Figure 1—figure supplement 6*).

The observation that cancer cells harboring deletion of both *TP53* and *FXR2* exhibited sensitivity to FXR1 inhibition suggested a collateral lethality correlation between FXR1 and *TP53* deletion. To investigate whether *TP53*/*FXR2* co-deletion is essential in determining cells' sensitivity to FXR1 inhibition, we monitored cell growth rate upon FXR1 downregulation in cell lines carrying homozygous deletion of either *TP53* (H1299, L540, MG-63, SKOV3) or *FXR2* (Hep3B) alone. The levels of p53, FXR2 and FXR1 were confirmed by q-RT-PCR and WB (*Figure 1—figure supplement 2*). In the tested cell lines, the proliferation rate was unaltered (*Figure 1F*), suggesting that concomitant

deletion of *FXR2* and *TP53* is necessary for triggering FXR1 inhibition-induced cell lethality. The role of FXR1 in controlling cell proliferation in *TP53/FXR2* co-deleted cancer cells was further confirmed in an *in vivo* xenograft. Dox-induced FXR1 downregulation significantly reduced tumor growth in the *TP53/FXR2* co-deleted cell line HL-60-derived xenograft model (*Figure 1G*), but had no impact on growth in the copy-number-normal cell A549-derived xenograft (*Figure 1—figure supplement 3C*).

To confirm the above observations, we engineered the copy-number-normal cell line AGS that is suitable for use with CRISPR (clustered regularly interspaced short palindromic repeat)/Cas9 to generate the isogenic homozygous *TP53* and *FXR2* double knockout (DKO) cell clone, a *TP53* single KO (*TP53* KO) cell clone, and a *FXR2* single KO (*FXR2* KO) cell clone. The knockout strategy is illustrated in *Figure 2—figure supplement 1*. Both the mRNA and the protein levels of p53 and FXR2 in the individual knockout clones were determined by WB (*Figure 2A*) and q-RT-PCR (*Figure 2—figure supplement 2A*). Consistent with the phenotypes observed in cancer cell lines, shRNA-induced FXR1 downregulation significantly inhibited cell proliferation only in the *TP53* and *FXR2* double knockout clones (*Figure 2B*) and had no effect on the *TP53* single KO (*Figure 2C*) or *FXR2* single KO (*Figure 2D*) clones. The FXR1-inhibition-induced anti-proliferative activity observed in *TP53/FXR2* double knockout cells was confirmed in the Matrigel assay (*Figure 2—figure supplement 2B*). Resistance to FXR1 inhibition in both *FXR2* single knockout clones (*Figure 2D*) and the *FXR2* homozygous deletion cell line Hep3B (*Figure 1E*) suggested that p53 deficiency is required for FXR1 to function in regulating cell proliferation. We downregulated p53 using RNAi in *FXR2* KO cells that stably expressed FXR1 shRNA to further investigate the necessity for p53 deficiency in FXR1's function in proliferation control. In line with its tumor suppressor function, *TP53* knockdown by siRNA enhanced cell growth in the tested cells (*Figure 2—figure supplement 2C*). Interestingly, *TP53* knockdown granted FXR1 the capability to regulate cell proliferation: diminished FXR1 suppressed the growth of *TP53* siRNA-treated *FXR2* KO cells but not of control siRNA-treated *FXR2* KO cells (*Figure 2—figure supplement 2C*). Moreover, knockout of *TP53* and *FXR2* didn't change the expression of FXR1 or FMR1 (*Figure 2—figure supplement 2D*). To confirm the redundant function shared by FXR1 and FXR2, we also used the same system to engineer a *TP53/FXR1* DKO clone and stably expressed FXR2 shRNA-2 or FXR2 shRNA-3. We observed that knockdown of FXR2 upon Dox induction suppressed proliferation (*Figure 2—figure supplement 3*), further confirming the redundancy between FXR1 and FXR2.

*Table 1* summarizes the studies in the cancer cell lines and CRISPR-Cas9 knockout clones. Using an arbitrary 30% proliferation inhibition as a cut off, there was a clear pattern of FXR1 regulation of cell proliferation in *TP53* and *FXR2* co-deleted cancer cells. Therefore, our data validate the role of FXR1 inhibition in blocking cell proliferation in *TP53*-deleted cancers in a collateral lethality manner upon passenger deletion of *FXR2*.

## The tandem Tudor domain is required for FXR1 activity in cell proliferation regulation

FXR1 contains multiple functional domains including the newly discovered N-terminal tandem Tudor domain, KH domains and the RGG box, as illustrated in *Figure 3A*. In order to pinpoint the domain responsible for regulating cell proliferation, we examined the ability of FXR1 isoforms to rescue FXR1-inhibition-induced cell growth reduction. There are three major isoforms, including full-length FXR1_a, C-tail truncation FXR1_b, and N-terminal tandem Tudor truncation FXR1_c (isoform gene codes are listed in Materials and methods). Ectopic expression of the shRNA-resistant plasmids encoding the three isoforms restored the expression of individual proteins in cells depleted of endogenousFXR1 (*Figure 3B*). Interestingly, only the full-length (FXR1_a) could rescue FXR1 shRNA-induced anti-proliferation. The Tudor-domain-truncated version FXR1_c completely lost its ability to rescue cell proliferation (*Figure 3B*). The C-tail truncation FXR1-b only partially rescued cell proliferation (data not shown). These data indicate that the tandem Tudor domain is necessary for FXR1 to regulate cell proliferation.

It has recently been reported that the tandem Tudor domain of FMR1 can recognize methylated lysine at histone H3 and recruit DNA repair protein, thus playing a role in DNA damage repair (*Alpatov et al., 2014*). We used the histone methyl lysine analog (MLA) protein pulldown assay to examine whether histones are bound to the FXR1's tandem Tudor domain. The Tudor domain showed binding to histone H3 containing methylated lysine at various positions, especially H3K4me1, H3K4me3, H3K9me3, H3K27me2, H3K36me1, and H3K79me2 (*Figure 3C*). The histone

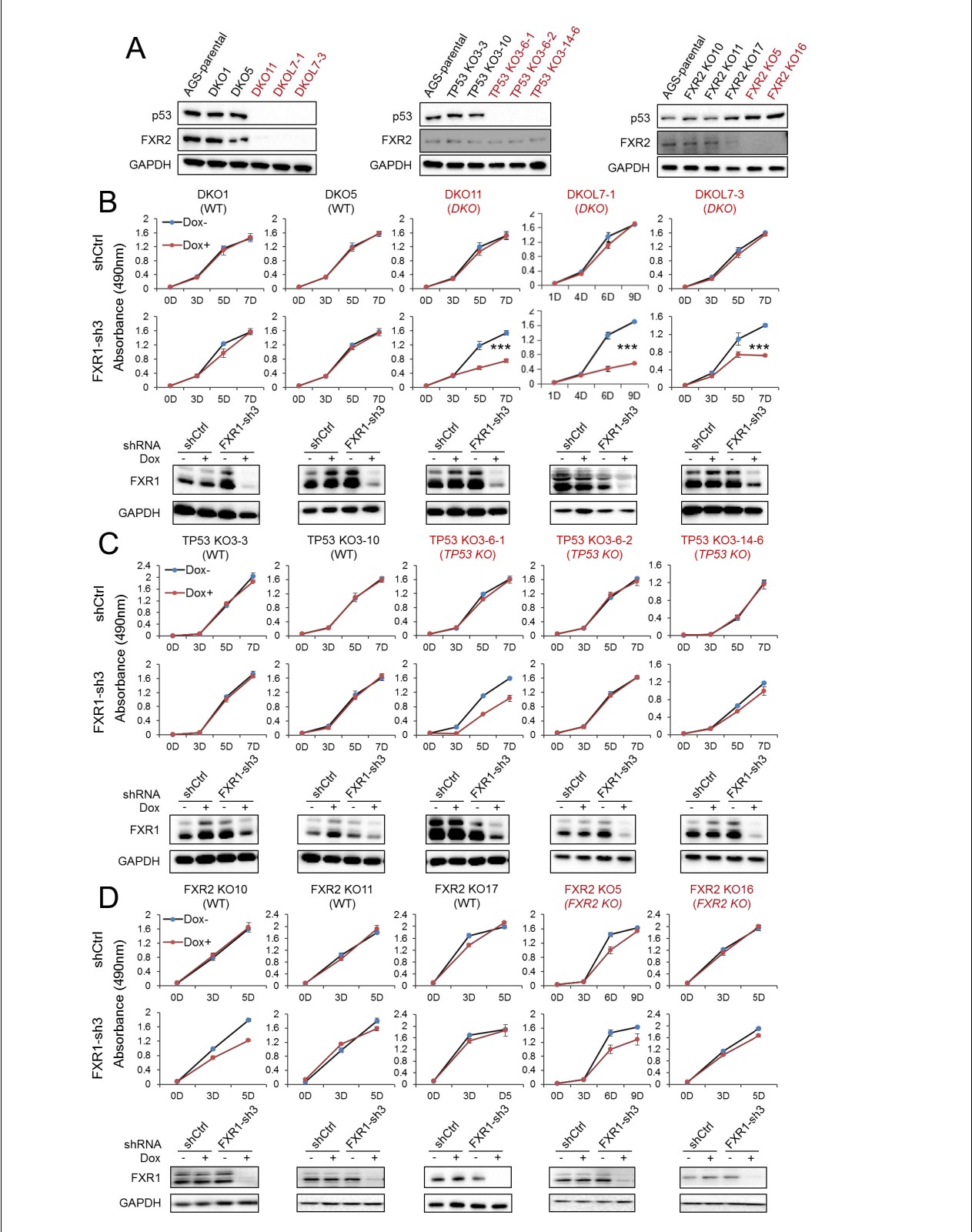

**Figure 2.** FXR1 knockdown inhibits cell proliferation in engineered *TP53/FXR2* double knockout cancer cell clones. (**A**) Protein levels of p53 and FXR2 in the CRISPR-Cas9-engineered copy-number-normal (WT) and knockout (KO) cell clones generated from the *TP53/FXR2* copy-number-normal parental cancer cell line AGS. Western blots (WBs) for individual cell clones from *TP53/FXR2* double knockout (DKO) (left), *TP53* single knockout (*TP53* KO) (middle), and *FXR2* single knockout (*FXR2* KO) lines (right) were analyzed in the presence of doxorubicin treatment (1 μM for 24 hr to trigger p53

*Figure 2 continued on next page*

*Figure 2 continued*

expression). Wild type clones are in black, and KO clones are in red. (**B–D**) Cell proliferation in the engineered AGS cell clones upon FXR1 knockdown. Upper panel, measurement of cell growth using the MTS assay. Lower panel, Dox-induced FXR1 knockdown in WB assay. Data represent the mean ± s.d. of three independent experiments. The cell proliferation rate was determined by measuring absorbance at 490 nm in an MTS assay (Y axis). *p<0.05, **p<0.01, ***p<0.001. Also see *Figure 2—figure supplements 1*, *2* and *3*.

DOI: https://doi.org/10.7554/eLife.26129.017

The following source data and figure supplements are available for figure 2:

**Source data 1.** Source data for *Figure 2*.
DOI: https://doi.org/10.7554/eLife.26129.021
**Figure supplement 1.** CRISPR-Cas9-engineered *TP53/FXR2* knockout clones.
DOI: https://doi.org/10.7554/eLife.26129.018
**Figure supplement 2.** Effect of FXR1 knockdown on cell proliferation in CRISPR-Cas9-engineered clones.
DOI: https://doi.org/10.7554/eLife.26129.019
**Figure Supplement 2—source data 1.** Source data for *Figure 2—figure supplement 2*.
DOI: https://doi.org/10.7554/eLife.26129.022
**Figure Supplement 3.** FXR2 knockdown inhibits cell proliferation in engineered *TP53/FXR1* double-knockout cancer cell clones.
DOI: https://doi.org/10.7554/eLife.26129.020
**Figure supplement 3—source data 1.** Source data for *Figure 2—figure supplement 3*.
DOI: https://doi.org/10.7554/eLife.26129.023

binding pattern of the tandem Tudor domain of FXR1 is similar to that of FMR1. These data indicate that the tandem Tudor domain can potentially bring FXR1 to the chromatin interface, thus playing a role in proliferation control.

## FXR1 locates at gene promoters and regulates transcription

We then investigated how FXR1 regulates proliferation in *TP53/FXR2* co-deleted cancer cells. In order to obtain a comprehensive assessment of FXR1's function, we conducted chromatin immuno-precipitation using a validated FXR1-specific antibody, followed by mass spectrometry (ChIP-MS), aiming to capture both the chromatin-associated and chromatin-free protein complexes in the cells. Proteins identified in the FXR1 pull-down complex from both H358 and KATOIII cells were analyzed. They are listed in *Supplementary file 1* and exemplified in *Figure 4A*. The enrichment analysis using the Database for Annotation, Visualization and Integrated Discovery (DAVID) clustered the potential FXR1-interacting proteins according to their function (*Supplementary file 2*). Proteins involved in mRNA binding, stability, splicing/metabolism, transportation, and translation were most enriched (*Figure 4b*, *Supplementary file 2*), consistent with the known function of FXR1. Interestingly, proteins participating in chromatin/chromosome organization and transcription signaling were also enriched (*Figure 4B* and *Supplementary file 2*). Next, we confirmed the protein interactions using ChIP followed by WB (ChIP-WB). In agreement with our DAVID enrichment analysis, FXR1 interacted with multiple proteins that are involved in chromatin and transcriptional signaling, including the transcription factors STAT1 and STAT3, the chromatin regulator CHD4, SNF2H, histone H2B and H3K4me3, and DNA topoisomerase TOP2A in H358 cells (*Figure 4C*). We discovered that phosphorylated STAT1 or STAT3 can also interact with FXR1 from the ChIP-WB analysis in the same cells (*Figure 4C*, lower panel). Importantly, the in vitro pull-down assay using the purified tagged proteins showed that FXR1 may directly interact with both STAT1 and STAT3 (*Figure 4D*). Interestingly, FXR1 had no impact on the phosphorylation of STAT1 or STAT3 or on their shuttling from the cytoplasm to the nucleus (*Figure 4—figure supplement 1*). When combined together evidence of FXR1's capacity to bind methylated histone H3, these results suggest that FXR1 may be involved in chromatin signaling and gene transcription.

To confirm the role of FXR1, we conducted chromatin immunoprecipitation coupled with high-throughput sequencing (ChIP-seq) in the *TP53/FXR2* co-deleted H358 cells using FXR1-, STAT1-, STAT3-, H3K4me3-, H3K9me3-, or H3K27me3-specific antibodies. We identified unfiltered 882 peaks of FXR1 (*Supplementary file 3*), primarily located at gene promoters close to transcriptional start sites (TSS) (*Figure 5A–B* and *Supplementary file 3*). In agreement with our hypothesis, STAT1, STAT3 and H3K4me3 were also primarily located at TSS (*Figure 5B* and *Figure 5—figure supplement 1A* and *Supplementary file 3*). Interestingly, the location of FXR1 significantly overlapped

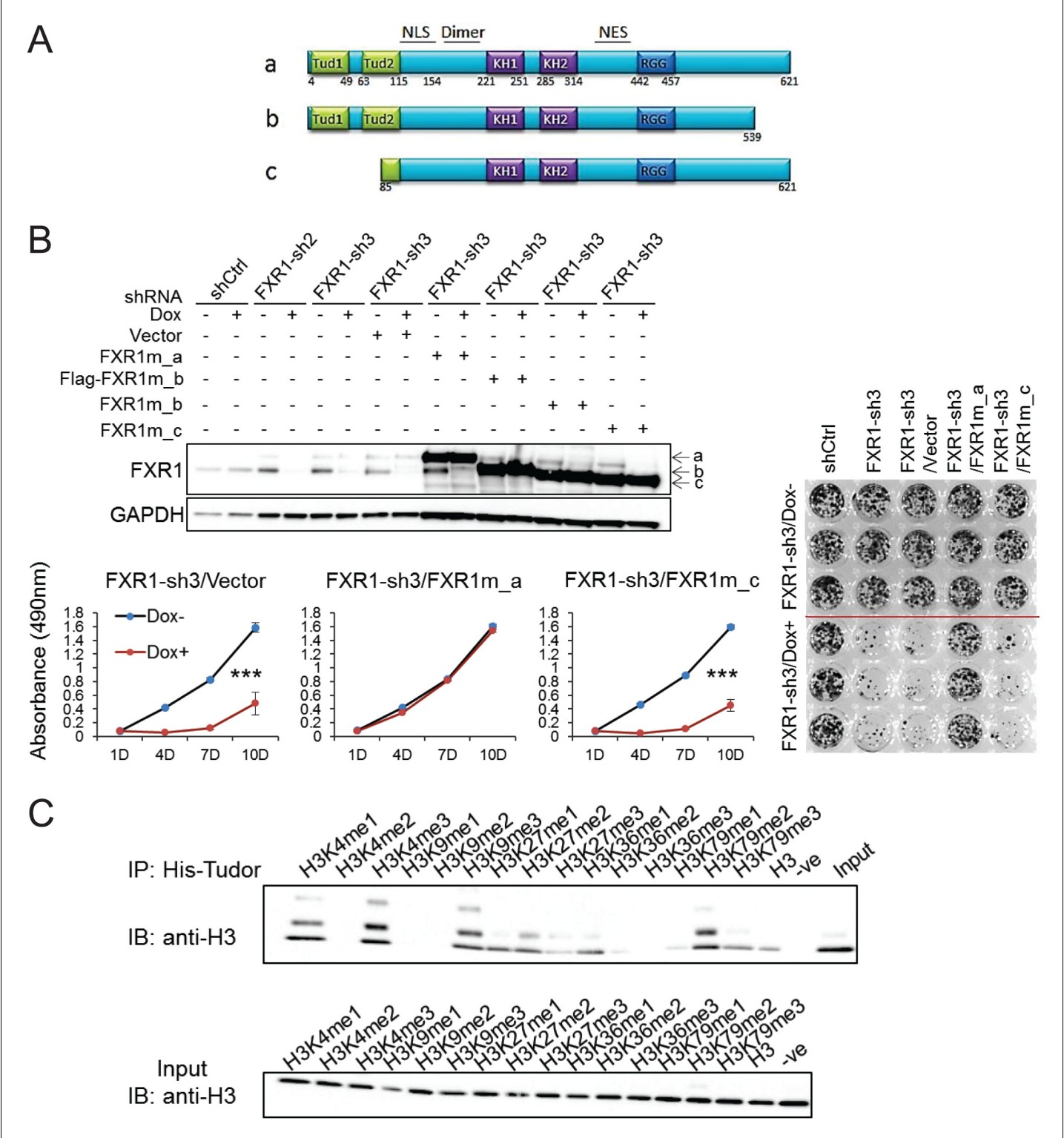

**Figure 3.** Tandem Tudor domain is required for FXR1's function in cell proliferation. (**A**) Schematic diagrams of FXR1 domains and isoforms: the full-length isoform a, the C-terminal tail truncated isoform b, and the N-terminal tandem Tudor truncated isoform c. (**B**) Comparison of rescuing capability among FXR1 isoforms in the H358 cell line. Upper, ectopic expression of the FXR1 shRNA-resistant isoforms (FXR1m_a, FXR1m_b, Flag-tagged-FXR1m_b, and FXR1m_c) upon Dox-induced FXR1 knockdown in a WB assay. Lower, cell proliferation measurement by MTS assay. Data represent the mean ± s.d. of three independent experiments. The crystal violet cell-staining represents the duplicate study at the last time point (after 10 days of Dox treatment). The cell proliferation rate was determined by measuring absorbance at 490 nm in an MTS assay (Y axis). *p<0.05, **p<0.01, ***p<0.001. (**C**)

*Figure 3 continued on next page*

*Figure 3 continued*

Tudor-histone H3 interaction assessed using the methyl lysine analog (MLA) protein with lysine methylation at various residues and His-tag FXR1 tandem Tudor protein in an in vitro pull down assay. –ve, negative control, pulldown of H3K4me3 MLA protein without the His-tagged-Tudor$_{FXR1}$.

DOI: https://doi.org/10.7554/eLife.26129.024

The following source data is available for figure 3:

**Source data 1.** Source data for *Figure 3*.

DOI: https://doi.org/10.7554/eLife.26129.025

with those of STAT1, STAT3, and H3K4me3 at their target genes (*Figure 5C* and *Supplementary file 3*). The percentage of overlapping increased when we focused on TSS (*Figure 5C* and *Supplementary file 3*). The FXR1-H3K4me3- or FXR1-STAT1/3-overlapped peaks primarily located in TSS regions, whereas the non-overlapped peaks primarily located in intergenic regions (*Figure 5—figure supplement 1B* and *Supplementary file 3*). At TSS, a total of 210 peak-associated genes were identified for FXR1 (*Figure 5C* and *Supplementary file 3*). Among them, 145, 147 and 200 peak-associated genes overlapped with STAT1, STAT3 and H3K4me3, respectively (*Figure 5C* and *Supplementary file 3*). FXR1 peaks rarely overlap with H3K9me3 or H3K27me3 (for examples, see *Figure 5D*, *Figure 5—figure supplement 2*, and *Supplementary file 3*), indicating FXR1's potential role in facilitating active transcription. Peaks aligning FXR1, STAT1, STAT3 and selective histone markers of representative genes — *CASC4*, *ARID1A*, *GLI1*, *SSBP2*, *CITED1*, *NPAS3* and *TRAPPC9* — are depicted in *Figure 5D* and *Figure 5—figure supplement 2*.

We selected FXR1-peak-associated putative target genes (239 genes listed in *Supplementary file 5*) and examined FXR1's role in their expression using q-RT-PCR. Upon inducible FXR1 knockdown, genes *ARID1A*, *CASC4*, *TRAPPC9*, *GLI1*, *CITED1*, *NPAS3*, and *SSBP2* were consistently downregulated in both H358 and KATOIII cells. Data from H358 cells are shown in *Figure 5E*. Using ChIP-PCR, we further investigated whether FXR1 directly occupies the promoter of these target genes. On the basis of the ChIP-seq peak position, we designed and selected two PCR primer pairs against each target gene's promoter region (*Figure 5—figure supplement 2B*) to capture FXR1's localization in the ChIP-PCR assay. The promoter regions of the target gene were detected in the FXR1 ChIP pull-down complex, indicating that FXR1 is located at the target gene promoters (*Figure 5F*, shCtrl/Dox- and FXR1-sh3/Dox-). Dox treatment did not reduce FXR1's occupancy in the shRNA control cells (shCtrl/Dox+, *Figure 5F*), but it evicted FXR1 from the gene promoter in FXR1-shRNA-expressing H358 cells (FXR1-sh3/Dox+, *Figure 5F*), confirming the specificity of the FXR1 ChIP assay. Using the same approach, STAT1 or STAT3 were located in the vicinity of FXR1 at the target gene promoter (*Figure 5F*). More interestingly, FXR1 knockdown resulted in STAT1 and STAT3 eviction from the gene promoters (*Figure 5F*), suggesting that FXR1 is required to recruit or stabilize STAT1 and STAT3 at gene promoters. On the other hand, STAT3 did not affect FXR1's occupancy of sites at gene promoters because its knockdown had no effect on the localization of FXR1 at promoters (*Figure 5G*, *Figure 5—figure supplement 3*).

To further confirm the redundant function shared between FXR1 and FXR2 as well as to investigate the interaction between FXR1/2 and STAT1/3, we conducted FXR1, FXR2, STAT1, STAT3, and H3K4me3 ChIP-seq in AGS cells. The data indicated that the four proteins were enriched at TSS regions in the genome (*Figure 5H*, *Figure 5—figure supplement 4A* and *Supplementary file 3*). FXR1 peak-associated genes had 89.95% (2935/3263 FXR1-peak-associated genes) overlap with FXR2 peak-associated genes in AGS cells (*Figure 5I*), and this percentage increased when we focused on TSS (*Figure 5—figure supplement 4B* and *Supplementary file 3*), suggesting a shared function of FXR1 and FXR2 at gene promoters. Only a small percentage of FXR2 binding-site-associated or peak-associated genes overlapped with those similarly associated with FXR1 in AGS cells. The difference is probably due to the weaker capability of the FXR1 ChIP antibody in pull down assay when compared to the FXR2 ChIP antibody, as reflected by the difference in the peak-associated gene numbers for FXR1 and FXR2 (3263 and 15,346), respectively (*Figure 5I*). There was also significant overlapbetween FXR1-associated genes and STAT1/3-associated genes, suggesting at least partially, that FXR1 mediates STAT1/3 transcriptional activity in AGS cells (*Figure 5I*, *Figure 5—figure supplement 4B*). The ChIP-seq peaks at the promoter region of the representative gene *ARID1A* in AGS cells are shown in *Figure 5—figure supplement 4C*. Our ChIP-PCR results revealed

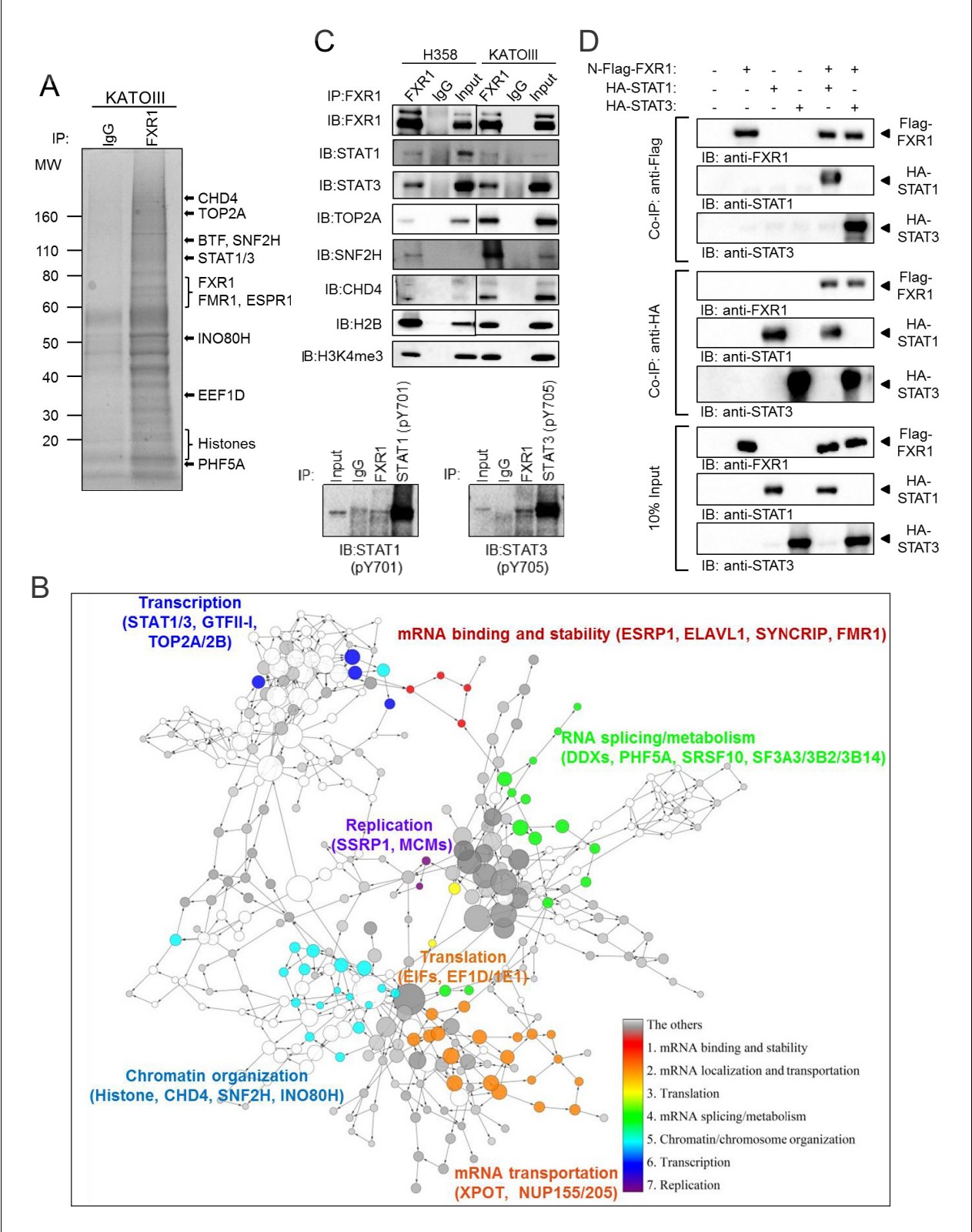

**Figure 4.** FXR1 protein complex identified by chromatin immunoprecipitation–mass spectrometry. (**A**) Coomassie-blue-stained sodium dodecyl sulfate (SDS) gel showing the endogenous proteins immunoprecipated from cross-linked and single-nucleosome-generated KATOIII cell lysate using validated anti-FXR1 antibody or IgG control. The potential FXR1 interacting proteins identified by the ChIP-MS assay followed by Cytoscape/GenePro analysis are labelled listed on the right side according to their molecular weight (MW). Also see ***Supplementary file 1***. (**B**) FXR1 protein complexes identified by

*Figure 4 continued on next page*

*Figure 4 continued*

ChIP-MS in H358 and KATOIII from Cytoscape/GenePro screenshot. Each circle represents an individual FXR1-interacting protein. Circle size indicates Gene Ontology (GO) term scale. The lines connecting the circles indicate the protein–protein interactions analyzed using the Kyoto Encyclopedia of Genes and Genomes (KEGG) pathway database. The Database for Annotation, Visualization and Integrated Discovery (DAVID) was used to cluster the potential FXR1-interacting proteins according to their function. Different colors represent selected major protein clusters, including mRNA binding and stability (red), mRNA localization and transportation (orange), translation (yellow), mRNA splicing/metabolism (green), chromatin/chromosome organization (light blue), transcription (dark blue), and replication (purple). Representative proteins are also listed for each function cluster. Also see *Supplementary file 2*. (C) Validation of the interaction between FXR1 and the proteins identified by ChIP-MS using the ChIP-Western blot (WB) assay by analyzing ChIP pull-down complexes in SDS-PAGE gel with validated specific antibodies. The lower panel shows the result in the H358 cell line. (D) The interaction between FXR1 and STAT1 or STAT3 evaluated by in vitro pull-down assay using purified Flag-tagged-FXR1 (full-length isoform a) and HA-tagged-STAT1/3. Anti-Flag or anti-HA antibodies were used to pull down the purified proteins. The proteins are detected using their specific antibodies in an immunoblotting assay. Also see *Figure 4—figure supplement 1*.
DOI: https://doi.org/10.7554/eLife.26129.026
The following figure supplement is available for figure 4:

**Figure supplement 1.** FXR1 knockdown does not affect the cytoplasm-nucleus shuttling or phosphorylation of STATs.
DOI: https://doi.org/10.7554/eLife.26129.027

a similar localization pattern for FXR2 as for FXR1, with both proteins occupying the promoters of target genes (*Figure 5J* upper panel and *Figure 5—figure supplement 4D*). Furthermore, the expression levels of target genes remained constant after FXR2 knockdown except for a slight inhibition of *CASC4* and *TRAPPC9* expression, further confirming the redundant function shared by FXR1 and FXR2 (*Figure 5J*, lower panel). Taken together, our data confirm that FXR1 is required to recruit or stabilize STATs on gene promoters, thus facilitating FXR1's transcriptional activity, and also show that FXR1 and FXR2 share similar function.

Our study showed that only a small portion of STAT1/3 peaks in the genome (0.7–1.57%) overlap with FXR1 peaks, indicating that FXR1 cooperates with STAT1/3 in a small number of STAT target genes' transcription in H358 cells (*Supplementary file 3*). By contrast, 24.38–25.74% of FXR1 peaks overlap with STAT1 and STAT3 peaks, suggesting a significant role for STATs in FXR1's transcriptional activity (*Supplementary file 3*). As STAT1 and STAT3 participate in FXR1 transcriptional activity, we hypothesized that a JAK inhibitor could repress FXR1 target gene expression. Indeed, we found that the JAK inhibitor S-Ruxolitinib suppressed STAT1 phosphorylation in a dose-dependent manner and inhibited the expression of some but not all FXR1 target genes in H358 cells (*Figure 5K*, *Figure 5—figure supplement 5*). Note that JAK inhibitor didn't suppress FXR1 expression (*Figure 5K*). These data confirm the involvement of STAT in FXR1's transcriptional activity.

We performed RNA-seq to analyze gene expression in H358 cells with biological triplicates. We selected FXR1-knockdown-induced gene expression changes using fold change $\log_2 > 0.6$ as the cutoff (*Supplementary file 6*). Among the seven FXR1 target genes identified by ChIP-seq, five showed significant downregulation upon FXR1 knockdown. The remaining two genes showed moderate downregulation (*Supplementary file 7*). Among the genes whose promoters are occupied by FXR1, 45 out of 210 genes showed regulation upon FXR1 knockdown in RNA-seq assay (*Figure 5—figure supplement 1D* and *Supplementary file 7*). RNA-seq revealed significantly more genes regulated by FXR1 than ChIP-seq. The discrepancy may be attribute primarily to the fact that FXR1 is a well-known mRNA-binding protein, regulating RNA metabolism and stability in addition to the newly discovered function in transcription regulation revealed in this study. Therefore, FXR1 knockdown could impact on both populations of genes, those regulated by FXR1's transcriptional activity and those regulated by FXR1's RNA-binding function. In addition, the scope and limitation of RNA-seq and ChIP-seq could also contribute to the discrepancy.

In order to assess whether target genes mediate FXR1's function in proliferation regulation, we utilized siRNAs to knockdown target genes and monitored the change in cell proliferation. The knockdown efficiency of the target genes by siRNAs was assessed using q-RT-PCR (*Figure 6—figure supplement 1A*). Among the seven target genes, knockdown of *GLI1* showed the most consistent anti-proliferative effect whereas *ARID1A*, *CITED1*, or *SSBP2* knockdown showed less inhibitory effect in a 7-day proliferation assay (*Figure 6—figure supplement 1B*). We further evaluated the combinatorial effect on cell proliferation of downregulation of both FXR1 and its target genes. We performed combinatorial partial knockdowns of both FXR1 and its target genes by applying lower concentration

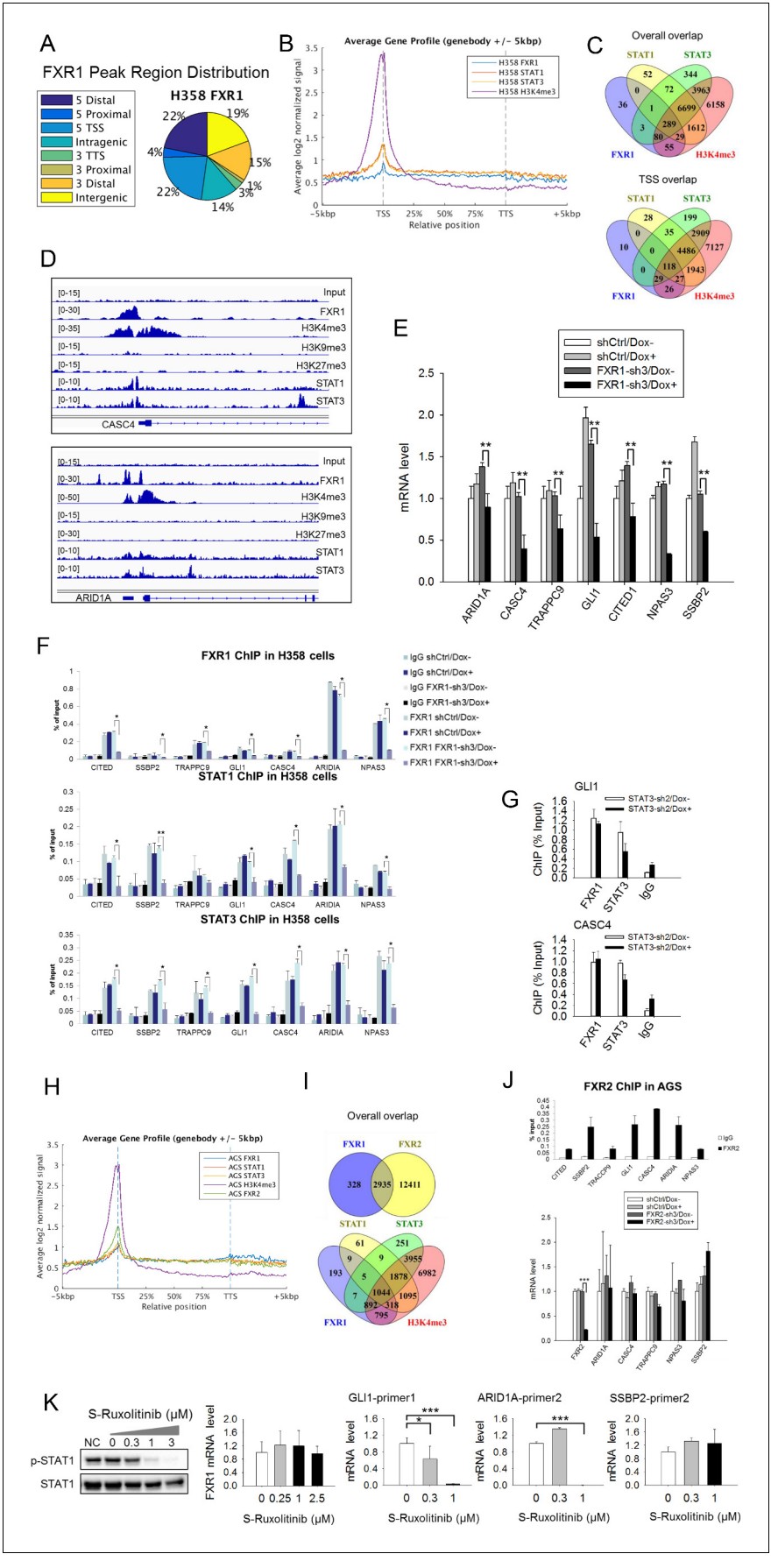

**Figure 5.** FXR1 regulates transcription by colocalizing with H3K4me3 and STAT1 and STAT3 at gene promoters. (**A**) Genomic distribution of FXR1 ChIP-seq peaks in H358 cells. 22% of peaks are enriched at promoter regions (defined as 5000 bp upstream and 1000 bp downstream of TSS) ($p<1\times e^{-5}$). (**B**) Average profile of FXR1, STAT1, STAT3, and H3K4me3 occupancies in gene bodies and promoters along the transcription units in H358 cells. The relative position is aligned by percentage from the transcription start site (TSS) to transcription terminal site (TTS). 5000 bp upstream of TSS and 5000 bp downstream of TTS are also included. (**C**) Upper: Venn diagram showing significant overlap of FXR1, STAT1, STAT3, and H3K4me3 ChIP-seq peak-associated genes in H358 cells (p<1×e-5). Lower: Venn diagram showing significant overlap of FXR1, STAT1, STAT3, and H3K4me3 ChIP-seq peak-associated genes (enriched at TSS) in H358 cells (p<1×e-5). (**D**) The enrichment of FXR1, H3K4me3, H3K9me3, H3K27me3, STAT1, and STAT3 ChIP-seq peaks at the gene promoter regions of *CASC4* and *ARID1A* from IGV screenshot in H358 cells. (**E**) Gene expression regulation by FXR1 knockdown in H358 cells using a quantitative real-time PCR (q-RT-PCR) assay. The mRNA level is presented as fold change normalized to GAPDH. The mRNA level of the shCtrl/Dox- sample is set as 1. \*\*p<0.01. (**F**) FXR1, STAT1 and STAT3 ChIP-PCR analyses at seven target genes in shCtrl and FXR1-sh3 H358 cell lines. Data are represented as mean ± s.d. of three independent experiments. IgG pull down is used as a negative control. \*p<0.05,\*\*p<0.01. (**G**) FXR1 and STAT3 ChIP-qPCR analyses at two selected genes in shCtrl and STAT3-sh2 H358 cell lines. Data are represented as mean ± s.d. of three independent experiments. IgG pull down is used as a negative control. \*\*\*p<0.001. (**H**) Average profile of FXR1, FXR2, STAT1, STAT3, and H3K4me3 occupancies at gene bodies and promoters along the transcription units in AGS cells. The relative position is aligned by percentage from the transcription start site (TSS) to transcription terminal site (TTS). 5000 bp upstream of TSS and 5000 bp downstream of TTS are also included. (**I**) Venn diagram showing significant overlap of FXR1 and FXR2 (upper), and FXR1, STAT1, STAT3, and H3K4me3 (lower) ChIP-seq peak-associated genes in AGS cells (p<1×e-5). (**J**) FXR2 regulation of FXR1 target-gene transcription. Upper, FXR2 ChIP-PCR analyses at seven selected genes in AGS cell lines. Data are represented as mean ± s.d. of three independent experiments. IgG pull down is used as a negative control. Lower, the inducible FXR2 knockdown and its effect on target-gene expression. The mRNA level is presented as fold change normalized to GAPDH. \*\*\*p<0.001. (**K**) JAK inhibitor effect on the expression of FXR1 target-gene expression in AGS cells. The JAK inhibitor (JAKi) S-Ruxolitinib inhibits STAT1 phosphorylation in a dose-dependent manner in H358 cells as measured by WB. The expression of both the target gene and FXR1 are determined by q-RT-PCR using their specific primers in S-Ruxolitinib-treated H358 cells. The mRNA level is fold-change normalized to GAPDH mRNA level. The mRNA level of 0 μM S-Ruxolitinib treatment sample is set as 1. \*p<0.05, \*\*\*p<0.001. Also see *Figure 5—figure supplements 1–6*, and *Supplementary files 3* and *4*.

DOI: https://doi.org/10.7554/eLife.26129.028

The following source data and figure supplements are available for figure 5:

**Source data 1.** Source data for *Figure 5*.
DOI: https://doi.org/10.7554/eLife.26129.035

**Figure supplement 1.** FXR1 colocalizes with H3K4me3 and STAT1/3 at gene promoters.
DOI: https://doi.org/10.7554/eLife.26129.029

**Figure supplement 2.** ChIP-seq peaks of FXR1, STAT1, STAT3 and histone marks (H3K4me3, H3K9me3, and H3K27me3) at validated target genes *GLI1*, *TRAPPC9*, *CITED1*, *NPAS3*, and *SSBP2*.
DOI: https://doi.org/10.7554/eLife.26129.030

**Figure supplement 3.** FXR1 localization at gene promoters is not affected by STAT3.
DOI: https://doi.org/10.7554/eLife.26129.031

**Figure Supplement 3—source data 1.** Source data for *Figure 5-Figure Supplement 3*.
DOI: https://doi.org/10.7554/eLife.26129.036

**Figure supplement 4.** FXR1 colocalizes with FXR2, STAT1/3 and H3K4me3 at gene promoters in AGS cells.
DOI: https://doi.org/10.7554/eLife.26129.032

**Figure supplement 4—source data 1.** Source data for *Figure 5-Figure Supplement 4*.
DOI: https://doi.org/10.7554/eLife.26129.037

**Figure supplement 5.** Regulation of target gene expression by JAK inhibitor.
DOI: https://doi.org/10.7554/eLife.26129.033

**Figure supplement 5—source data 1.** Source data for *Figure 5-Figure Supplement 5*.
DOI: https://doi.org/10.7554/eLife.26129.038

**Figure supplement 6.** Cell proliferation regulation by JAK inhibitor.
DOI: https://doi.org/10.7554/eLife.26129.034

**Figure supplement 6—source data 1.** Source data for *Figure 5-Figure Supplement 6*.
DOI: https://doi.org/10.7554/eLife.26129.039

of siRNA (5 nM) and shortening Dox treatment duration (to 4 days). As shown in the cell image and MTS detection (*Figure 6A*), the combined knockdown of both the target genes and FXR1 showed an additive effect on cell growth inhibition. Amongst the target genes, *GLI1* downregulation by siRNA showed the most dramatic combinatorial effect with FXR1 knockdown in suppressing cell proliferation.

Upon FXR1 inhibition, cells underwent blebbing, suggesting a role of FXR1 in cell apoptosis. Following this lead, we found that FXR1 knockdown induced the cleavage of PARP in H358 cells (*Figure 6B*), a signal event of cell apoptosis. Furthermore, FXR1 inhibition increased caspase 3/7 activity (*Figure 6B*). Consistently, knockdown of FXR1 target gene *GLI1* also induced apoptotic signals including the cleavage of PARP and activation of caspase 3/7 in the same cells (*Figure 6C*). We observed the similar effect of FXR1 shRNA in inducing PARP cleavage and caspase 3/7 activity in another cell line, HL-60 (*Figure 6—figure supplement 2*). More interestingly, the effect on PARP cleavage could be rescued by ectopic expression of shRNA-resistant full-length FXR1m_a, indicating a direct role of FXR1 in cell apoptosis signaling (*Figure 6—figure supplement 2*). These data suggest that downregulation of FXR1 inhibits cell growth through the activation of cell apoptotic signaling, at least partially through regulating target gene transcription.

Taken together, the findings of our study indicate that FXR1 inhibition suppresses cell proliferation in cancer cells containing *TP53* and *FXR2* homozygous deletions in a collateral lethality manner. FXR1 regulates gene transcription, at least in part, to enable control of cell proliferation.

## Discussion

The importance of p53 in cancer has led to tremendous efforts to target its activity by therapeutic approaches. For cancers containing normal *TP53* copy number, there has been good progress in interrupting MDM2–p53 interactions to stabilize and promote p53 tumor suppressor activity. For *TP53* mutated cancers, the current strategy is to restore p53 anti-tumor activity (*Khoo et al., 2014*; *Soragni et al., 2016*). No therapeutic strategies have been proposed to target *TP53* deletion cancers until a recent report demonstrated that *TP53* heterozygous deletion predisposes cancer cells to further suppression of *POLR2A*, a neighboring gene undergoing concomitant deletion (*Bradner, 2015*; *Errico, 2015*; *Liu et al., 2015*). However, this strategy will not be applicable to cancers with homozygous *TP53* deletions in which *POLR2A* is co-deleted in most cases (*Figure 1—figure supplement 1C*). Here, for the first time, we report an approach to target cancers with concomitant homozygous *TP53* and *FXR2* deletion by inhibiting FXR1. The passenger deletion of *FXR2* in the *TP53* deletion locus renders cancer vulnerable to FXR1 inhibition, which echoes with the recently raised concept of collateral lethality in tumor suppressor gene deletion (*Muller et al., 2015*; *Muller et al., 2012*; *Nijhawan et al., 2012*). In this study, several lines of evidence confirmed the redundant function between FXR1 and FXR2: 1) both FXR1 and FXR2 can rescue FXR1 shRNA-inhibited cell proliferation (*Figure 1D*), 2) FXR1 and FXR2 share a significant portion of overlapping binding sites in the genome (*Figure 5I*, *Figure 5—figure supplement 4B–C*), and 3) knockout of either protein sensitizes *TP53* homozygous deletion-containing cancer cells for the inhibition of the remaining member (*Figure 2D*, *Figure 2—figure supplement 2*). Although another family member, FMR1, shares structural and functional similarity with FXR1 and FXR2, our data suggest that FMR1 does not participate in the regulation of proliferation or in rescuing the FXR1 inhibition in the tested cell models. *TP53* homozygous deletion is observed in 1–15% of human tumors, the majority of which have *FXR2* homozygous deletions according to the TCGA database. This study therefore opens the avenue to the development of FXR1-based targeted therapies to treat *TP53* homozygous deletion cancers, in which there is a significant unmet medical need.

Interestingly, we have found that *TP53* deletion is required for cancer cells to respond to FXR1 inhibition. This was confirmed in our engineered *FXR2* single knockout cell clones which were only sensitive to FXR1 knockdown when p53 was downregulated. However, the mechanism of action remains elusive. There have been reports suggesting that p53 deficiency promotes the activation of JAK/STAT signaling, which may lead to cellular dependency on FXR1 because it regulates the localization of STAT1/3 at gene promoters (*Guo et al., 2014*; *Spehlmann et al., 2013*). Considering that FXR1 can both suppress p53 to escape senescence (*Majumder et al., 2016*) and activate gene transcription (evidence from this study), p53 deletion may augment FXR1's role in transcription regulation in controlling cancer cell proliferation. Given the various critical functions of p53 in governing

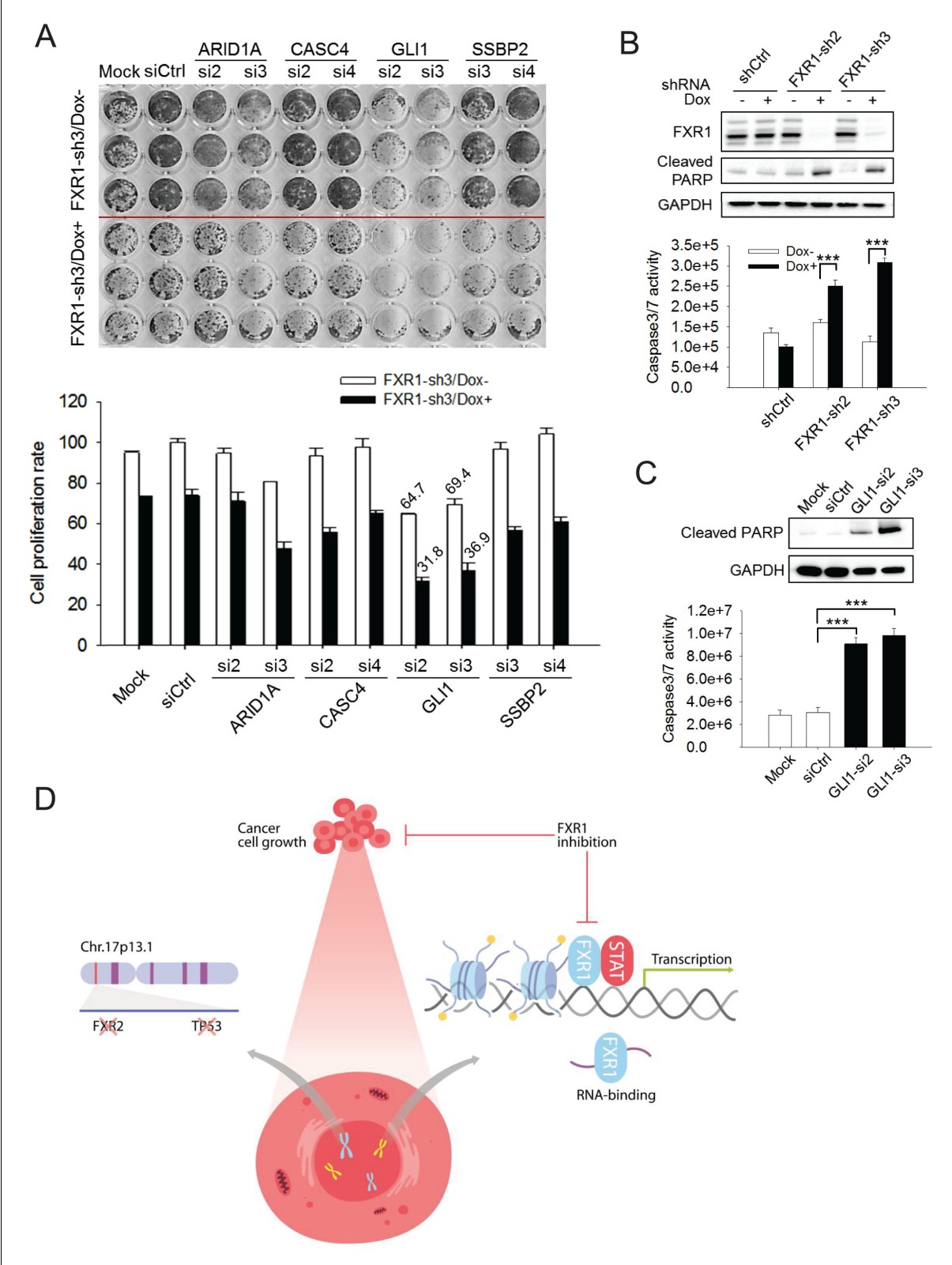

**Figure 6.** FXR1 target genes mediate its function in cell proliferation. (**A**) Dual knockdown of FXR1 and its target gene enhances cell growth inhibition. The H358/FXR1-sh3 stable cell line was transfected with target gene-specific siRNA (5 nM), and treated with Dox for induction of FXR1 knockdown one day post-transfection, and then subjected to cell imaging (upper) or MTS (lower) assay to monitor cell growth after four further days, after a total of 5 days of Dox treatment. Percentage of cell growth rate (absorbance at 490 nm) comparing to control siRNA (siCtrl) without Dox treatment group (rate

*Figure 6 continued on next page*

*Figure 6 continued*

set as 100) is labeled. Data represent mean ± s.d. of three independent experiments. (B) Cell apoptosis induced by FXR1 knockdown. Upper, cleaved PARP is measured by WB in H358/FXR1-sh2 and H358/FXR1-sh3 cells upon Dox-induced FXR1 knockdown for 5 days. Data represent one time study out of three independent experiments. Lower, caspase-3/7 activity is measured by Caspase-Glo3/7 assay in H358/FXR1-sh2 and H358/FXR1-sh3 cells. Data represent mean ± s.d. of three independent experiments. ***p<0.001 (FXR1-shRNA/Dox + sample comparing to FXR1-shRNA/Dox- sample). (C) Cell apoptosis induced by *GLI1* knockdown in H358 cells. Upper panel, cleaved PARP level detection. Data represent one time study out of three independent experiments. Lower panel, caspase-3/7 activity measurement. Data represent the mean ± s.d. of three independent experiments. ***p<0.001 (GLI1-shRNA/Dox + sample comparing to GLI1-shRNA/Dox- sample). (D) Schematic diagram demonstrating the cellular function and disease linkage of FXR1. In brief, FXR1 recognizes histone marks through its tandem Tudor domain, and recruits or stabilizes transcription factor STATs at target gene promoters for transcription regulation. Further, as an RNA-binding protein, FXR1 also functions in mRNA transport and the regulation of mRNA translation and stability. *TP53* homozygous deletion cancers usually have passenger deletion of the neighboring gene *FXR2* at chromosome 17p13.1, which causes cancer-specific cell dependency on the other FXR1 family member. Inhibition of FXR1 could serve as a novel therapeutic approach to targeting *TP53* homozygous deletion cancers that involve co-deletion of *FXR2* in a collateral lethality manner. Also see *Figure 6—figure supplements 1*, *2* and *3*.

DOI: https://doi.org/10.7554/eLife.26129.040

The following source data and figure supplements are available for figure 6:

**Source data 1.** Source data for *Figure 6*.
DOI: https://doi.org/10.7554/eLife.26129.044
**Figure supplement 1.** siRNA knockdown of FXR1 target genes.
DOI: https://doi.org/10.7554/eLife.26129.041
**Figure Supplement 1—source data 1.** Source data for *Figure 6-Figure Supplement 1*.
DOI: https://doi.org/10.7554/eLife.26129.045
**Figure supplement 2.** Cell apoptosis induced by FXR1 knockdown in HL-60 cells.
DOI: https://doi.org/10.7554/eLife.26129.042
**Figure supplement 2—source data 1.** Source data for *Figure 6-Figure Supplement 2*.
DOI: https://doi.org/10.7554/eLife.26129.046
**Figure supplement 3.** FXR1 binding to the target genes' mRNA.
DOI: https://doi.org/10.7554/eLife.26129.043

cellular behavior, its deficiency may create a cell dependency on FXR1 and FXR2 activities. The underlying molecular mechanism requires further investigation. Some studies have explored the oncogenic property of FXR1 in lung cancer with FXR1 overexpression or amplification (*Comtesse et al., 2007*; *Qian et al., 2015*). *FXR1* copy number gain occurs in some types of cancers, particularly in lung squamous carcinoma (*Comtesse et al., 2007*; *Qian et al., 2015*) and its occurrence is mutually exclusive with *TP53* and *FXR2* homozygous deletion according to TCGA database (*Figure 1A*). Together with our study, this evidence suggests that FXR1 controls tumor growth in cancers with either *TP53/FXR2* homozygous deletion or *FXR1* amplification.

Furthermore, we reveal a novel function of FXR1 in transcriptional regulation and elucidate its molecular mechanism in regulating cell proliferation, in addition to its well-established role in mRNA binding. Our ChIP-MS study revealed that FXR1 interacts with chromatin regulators and transcription factors. In addition, about a half of FXR1's peaks in the ChIP-seq assay co-localize with H3K4me3 peaks at gene promoters, suggesting a new role for FXR1 in facilitating active transcription. FXR1 interacts with and recruits or stabilizes STAT1 or STAT3 at the target gene promoters, suggesting that cooperation with STATs is required for its transcriptional activity. Indeed, the inhibitor of the STAT upstream kinase JAK suppressed some of FXR1's target genes expression. Reciprocally, FXR1 is required for the localization of STATs on its target gene promoters. Generally, it is believed that JAK-catalyzed STAT1/3 phosphorylation drives their shuttling to the nucleus and localization at gene promoters in order to regulate transcription (*Levy and Darnell, 2002*). Our findings advance the current views about the regulation of transcription by STAT1/3 by revealing the necessity for FXR1 in regulating STAT occupancy at promoters. STAT is well established in regulating cytokine gene transcription. We have also observed that FXR1 knockdown inhibited the expression of multiple cytokines (data not shown), consistent with a recent report of FXR1 function in positively regulating cytokine expression in monocytes (*Le Tonqueze et al., 2016*). Notably, only about 0.7–1.57% of the STAT1/3 peaks overlap with FXR1 peaks in our ChIP-seq study, suggesting that FXR1 only participates in the transcription of a very small fraction of JAK-STAT target genes. Although JAK inhibition suppressed the expression of some FXR1 target genes, it had no effect on inhibiting cancer cell

proliferation in FXR1 shRNA-sensitive cells (*Figure 5—figure supplement 6*). This is in line with the fact that JAK-STAT signaling pathway governs a much broader spectrum of gene expression and cellular behaviors. It w worth investigating whether other epigenetic regulators are involved in STAT transcriptional activity. Among the FXR1 peaks, approximately 25% overlapped with STAT peaks according to our study in H358 cells. FXR1 may also cooperate with other transcription factors in regulating transcription; examples include BTF (*Ma et al., 2014*) which was also identified in an FXR1 protein complex in our ChIP-MS study. In addition, we identified several other FXR1-interacting proteins that have essential function in transcription regulation, including TOP2A/2B, GTF2A, and FACT80. Further investigations are needed to dissect the underlying mechanism of FXR1 in transcription regulation.

FXR1 is an RNA-binding protein that regulates mRNA stability, transportation of RNA from the nucleus to the cytoplasm, and translation through its functional KH domains and RGG box. Our ChIP-MS data confirmed the established role of FXR1, which binds to and regulates mRNA processing and translation (*Figure 4B*). Furthermore, we confirmed ChIP-MS findings and discovered that FXR1 binds to the mRNA of target genes such as *CASC4*, *TRAPPC9*, *GLI1*, and *SSBP2* in the mRNA binding assays (*Figure 6—figure supplement 3*). Our study confirmed that, in addition to its direct function in transcription, FXR1 binds to some but not all of its target gene mRNAs, which may contribute to its role in proliferation regulation. This raises the possibility that FXR1 has dynamic functions in transcription and subsequent mRNA processing.

The newly discovered N-terminal tandem Tudor domain has a high degree of similarity among the three Tudor family members (*Adams-Cioaba et al., 2010*; *Hu et al., 2015*; *Myrick et al., 2015*). The Tudor domain in FMR1 was reported to bind to multiple methylated lysine at histone H3 and to participate in DNA damage repair (*Alpatov et al., 2014*). Our study shows that the tandem Tudor in FXR1 binds to methylated lysine in histone H3 in a similar manner as that in FMR1 and it is required for FXR1's function in regulating cancer cell proliferation. Further investigation is needed to study the amino acid residues and the structure of the aromatic cage within the FXR1 tandem Tudor domain that are responsible for histone binding, which might provide the foundation for development of Tudor domain inhibitors. The observation that FXR1 knockout mice die shortly after birth and knockdown of FXR1 causes abnormalities in striated muscle and heart in zebrafish (*Mientjes et al., 2004*; *Zarnescu and Gregorio, 2013*) raise potential safety concerns with regard to inhibiting FXR1 in cancer patients. However, the inducible FXR1 knockout in adult mice didn't display the lethal phenotype ( *Mientjes et al., 2004*), suggesting the potential of FXR1 as a drugging target. Overall, this study provides an important approach to targeting human cancers with *TP53* homozygous deletions. The novel function of FXR1 in transcription advances our knowledge regarding FXR1's function in gene expression on top of its known roles in mRNA processing and translation.

## Materials and methods

### TCGA and CCLE data analysis

Copy-number analysis in human tumor tissues was performed using data from The Cancer Genome Atlas (TCGA) (http://www.cbioportal.org/cross_cancer.do) as previously described (*Gao et al., 2013*). Gene copy numbers in human cancer cell lines were downloaded from the Cancer Cell Line Encyclopedia database (CCLE) (http://www.broadinstitute.org/ccle) and analyzed using an R package ComplexHeatmap (https://bioconductor.org/packages/release/bioc/html/ComplexHeatmap.html).

### Cell culture

KATOIII, HL-60, H358, MKN45, AGS, HepG2, A549, H1299, MG-63, SKOV3, Hep3B, and HEK 293 T cell lines were purchased from American Type Culture Collection (ATCC, Manassas, Virginia, USA, http://www.atcc.org). The KMS-11 cell line was obtained from Horizon Discovery Ltd. (Cambridge, UK). The L540 cell line was purchased from Biovector Science Lab, Inc (Beijing, China). Although L540 is listed as a misidentified cell line by ICLAC (International Cell Line Authentication Committee), we used it as a *TP53* single deletion cell model after comfirming the abscence of p53 mRNA and protein expression. The cell lines were cultured in RMPI 1640 medium (GIBCO) or DMEM medium

(GIBCO) supplemented with 10% fetal bovine serum (FBS, GIBCO) and 100 U/mL penicillin-strepto-mycin (GIBCO) at 37°C with 5% $CO_2$. All cell lines were authenticated by Short Tandem Repeat (STR) profiling and routinely monitored for mycoplasma contamination. The source and mycoplasma status as well as the Research Resource Identifiers (RRIDs) of each cell line are listed in *Supplementary file 8*.

## Cell proliferation and cell apoptosis

Cell proliferation rate was measured by MTS assay. Cell apoptosis was determined by analyzing Caspase3/7 activity. The luminescence signal in the Caspase3/7 assay was recorded by an Envision 2104 Multilabel reader (Perkin Elmer, Waltham, MA). The detailed procedure and information are in the Supplementary methods.

## Chromatin immunoprecipitation (ChIP)

ChIP sample preparation follows the manual of SimpleChIP Enzymatic Chromatin IP Kit (Magnetic Beads, CST, #9003). The experimental procedure and reagent information for ChIP-WB, ChIP-MS, ChIP-seq, or ChIP-PCR are in the Supplementary methods.

## Statistical analysis

In cell assays, three independent replicates were included in each experiment. Student T Test was performed to compare the differences between two groups with normal distribution. All data were analyzed using GraphPad Prism 6.

## Short hairpin RNA (shRNA) construction and stable cell generation

Gene-specific shRNA sequences were obtained from the Sigma website (http://www.sigmaaldrich.com/), and the hairpin sequences were cloned into the doxycycline (Dox)-inducible lentiviral vector pLKO-Tet-On with Puromycinas as a selection marker. Five shRNAs targeting each gene (*FXR1*, *FXR2* and *STAT3*) were screened.

Lentiviral particles were packaged in HEK 293 T cells by transient transfection. Briefly, 6 µg of shRNA encoding lentiviral plasmid, 0.6 µg of helper PCG10 plasmid and 5.4 µg of helper PCG41 plasmid were mixed with 30 µl X-tremeGENE HP DNA Transfection Reagent (Roche, 06366236001) and co-transfected into HEK 293 T cells at 70–80% confluency in a 10 cm Petri dish. At 72 hr post transfection, the supernatant containing virus particles was collected and filtered through a 0.45 µm PVDF Millex syringe filter.

Each cell line was subjected to a puromycin tolerance test to determine the optimal concentration for establishing stable cell lines. To generate a shRNA-expressing stable cell line, cells were infected with lentiviral particles with 8 µg/ml polybrene. After 48 hr, the cells were selected by 0.5–2 µg/ml puromycin (GIBCO, A11138-03) for 7 days. shRNA expression was induced by 0.1–1 µg/ml Dox (Sigma, D9891), and the knockdown efficiency was tested by WB assay or q-RT-PCR. Dox concentration was optimized for each stable cell line to achieve the most optimal knockdown efficiency at the minimum dose.

For ectopic expression of proteins of interest, cells were infected with the lentiviral pLenti6.3-MCS construct encoding the *FXR1, FXR2* or *FMR1* gene open reading frame and selected using 2–6 µg/ml blasticidin (GIBCO, A11139-03) for stable cell line generation. The inserted open reading frames are: FXR1_a, NM_005087.3; FXR1_b, NM_001013438.2; FXR1_c, NM_001013439.2; FXR2, – NM_004860.3; and FMR1, NM_002024.5. The sequence of the FXR1 shRNA resistant mutation was CGCCAGGTTCCATTTAATGAA mutated to CGTCAAGTACCGTTCAACGAG and TTGCGAAGTATTCGTACGAAGTTG mutated to TTACGGAGCATCCGAACAAAATTA.

FXR1 shRNAs were inserted in adenoviral vector pAd/CMV/IRES/GFP. The adenoviruses were produced in HEK 293 T cells by transient transfection of the plasmids. Virus titer was determined by counting GFP-positive cells after adenoviral infection. In the proliferation assay, adenoviruses were added to the cell culture medium and replenished 4 days afterwards to achieve efficient FXR1 knockdown.

The shRNA target sequence is listed in *Supplementary file 8*.

## Western blot analysis

Cells were lysed using RIPA lysis buffer (CST, 9806S) with complete protease inhibitor cocktail (Sigma, P8340) and phosphatase inhibitor cocktail (Sigma, P5726). The protein concentration was detected by BCA kit (Thermo Scientific, Cat#23225). The protein lysate was denatured by adding NuPAGE sample reducing agent (Novex, Invitrogen, NP0009) and NuPAGE LDS Sample buffer (Novex, Invitrogen, NP0007) followed by heating at 95°C for 5 min. Equal amounts of protein lysate was loaded to the NuPAGE Electrophoresis System (Life Technologies) according to the manufacturer's protocol. Proteins in the gel were transferred to the nitrocellulose membrane in iBlot 2 Transfer Stacks (IB23001 and IB23002) using the iBlot 2 Dry Blotting System (Life Technology, IB21001). The membrane was blocked in SuperBlock Blocking Buffer in TBS (Pierce, 37535), incubated with the specific primary antibodies for the protein of followed by reaction with the secondary antibodies. The protein signals were detected using SuperSignal West Femto Maximum Sensitivity Substrate (Thermo, 34095), and were scanned and quantified using ChemiDoc MP System (BIO-RAD).

The antibodies and their Research Resource Identifiers (RRIDs) used in this study are listed in *Supplementary file 8*.

## Cell proliferation and cell apoptosis

Cell proliferation rate was detected by MTS assay. Briefly, the cells were seeded in 96-well plates at optimal density in 100 µl medium per well followed by treatment. Cells were added with CellTiter 96 AQueous One Solution Reagent (Promega, G3581) at the indicated time by following the manual instructions and subjected to absorbance recording at 490 nm using a PowerWave HT Microplate Spectrophotometer (BioTek). The cell proliferation rate in MTS was determined by measuring absorbance at 490 nm. In the duplicated experiment, cell images were scanned after fixing the cells with cold methanol and staining with 0.1% crystal violet. In some cases, the cells were seeded on the solidified Matrigel (BD Bioscience) layer in 96-well plate (50 µl per well) to measure the organoid growth.

Cell apoptosis was determined by analyzing Caspase3/7 activity using the Caspase-Glo3/7 assay kit (Promega, G8092) according to the kit instructions or by detecting the protein level of cleaved PARP (antibody from Cell Signaling Technology, 9541) in Western blot. The luminescence signal in the Caspase3/7 assay was recorded by an Envision 2104 Multilabel reader (Perkin Elmer).

## Tumor xenograft model

The HL-60-derived xenograft model in female BALB/c nude mice was used to determine FXR1's function in tumor growth. Briefly, the female BALB/c nude mice (age 6–8 weeks) housed under pathogen-free conditions were inoculated subcutaneously in the right flank with HL-60 cells stably expressing control shRNA (shCtrl) or Doxycycline-inducible FXR1-sh3. Each mouse received $5 \times 10^6$ cells in 0.2 ml PBS:Matrigel (1:1 mixture) for tumor development. When tumor size reached approximately 180 mm$^3$ on day 14, mice were randomly grouped (n = 6) and Doxycycline (2 mg/ml) was added to drinking water. Doxycycline was replenished twice per week. Tumor volume and body weight were measured twice a week. Tumor size was measured using a calliper, and tumor volume was calculated according to the following equation: tumor volume (mm$^3$) = length (mm) × width$^2$ (mm$^2$) × 0.5. The experiment was terminated when the tumor volume reached 2,000 mm$^3$ according to animal welfare guidelines. All animal studies were carried out under IACUC number R20150728-Mouse and Rat approved by the Institutional Animal Care and Use Committee (IACUC) of WuXi AppTec. Summary statistics, including mean and the standard error of the mean (s.e.m), were provided for the tumor volume of each group at each time point. The A549-derived xenograft model was studied using a similar method.

## CRISPR-Cas9 genomic editing system to generate knockout cells

Guide RNA (gRNA) was designed according to the formula of GN20GG or CCN20C in the forward or reverse strand of genomic DNA, respectively (*Cong et al., 2013*; *Ran et al., 2013*; *Shalem et al., 2014*; *Wang et al., 2013*). Two sets of gRNAs targeting the 5' and 3' sequences of the genomic region of interest(*TP53*-to-*FXR2*, *TP53* alone, *FXR1* alone, or *FXR2* alone)were designed. The synthesized double-strand DNA of the gRNA was inserted into the pU6-gRNA_SPycas9-2Acherry vector expressing Cas9 protein (*Mavrakis et al., 2016*). To test gRNA efficiency, gRNA-expressing

plasmids were transfected into cells using XtremeGene reagent (Roche). Genomic DNA was extracted using the Purelink Genome DNA mini kit (Invitrogen, K1820-02). PCR was performed using genomic DNA as the template and the Accuprime Pfx SuperMix PCR system (Invitrogen, 12344040) to amplify the sequence containing the gRNA PAM position, and PCR product was sequenced to determine the successful cut by gRNA-guided Cas9 by monitoring the miscellaneous peaks. The efficient gRNA target sequences were: TP53-5'-gRNA, GGGCAGCTACGGTTTCCGTCTGG; TP53-3'-gRNA, GGTGTGCGTCAGAAGCACCCAGG; FXR2-5'-gRNA, GGAGGCGGTGGCGGCGCCATGGG; FXR2-3'-gRNA, GGGGGGTGGCACAGCAGCTTGGG; FXR1-5'-gRNA, GTGGAGGTTCGCGGCTC TAA; FXR1-3'-gRNA, GAAAAAAAAGTTGCTGGCTAT.

The donor plasmid contains the recombination sequences compatible with the 5' and 3' arms of the genomic region of interest and the GFP-coding sequence in vector pcDNA3.1. The mixture of gRNA-Cas9 plasmids targeting the 5' or 3' sequence of the genomic region and donor plasmid at ratio 1:1:1 was transfected into the candidate cells in a 10 cm Petri dish using XtremeGene. One week post-transfection, the GFP-expressing cells were sorted and seeded into a 96-well plate at a density of one cell per well using FACSArisII (BD). The single cell clones were cultured for two weeks to identify and filter out the cells with transient GFP signal. The clones with strong GFP signal, which would have potential insertion and replacement of the genomic region of interest by the donor sequence, were subjected to knockout validation by genomic PCR-sequencing, q-RT-PCR, and Western blot.

The sequence of genomic PCR primers is listed in *Supplementary file 8*.

## Chromatin immunoprecipitation (ChIP)

*ChIP sample preparation:* ChIP samples were prepared using SimpleChIP Enzymatic Chromatin IP Kit (Magnetic Beads, CST, #9003) according to the manufacturer's protocol. Briefly, cells cultured in 15 cm Petri dishes were crosslinked with 1% formaldehyde when reaching 70–80% confluence. Crosslinking was terminated by adding 10 × glycine followed by rinsing the cells twice with cold PBS, and $4 \times 10^7$ cells were collected into one 15 ml conical tube. The cell pellet was resuspended and incubated in Buffer A on ice for 10 mins for cell lysis. The nuclei were collected by centrifugation at 3,000 rpm for 5 mins at 4°C and resuspended in Buffer B followed by centrifugation and supernatant removal. The nuclei pellets were resuspended in 1.0 ml Buffer B, and transferred to a 1.5 ml microcentrifuge tube. 7.5 μl of Micrococcal Nuclease was added and incubated for 20 mins at 37°C to digest the DNA to lengths of approximately 150–300 bp. After the termination of digestion by adding 100 μl of 0.5 M EDTA, the nuclei pellets were collected by centrifuging at 13,000 rpm for 1 min at 4°C. The nuclear pellets were resuspended in 1 ml ChIP buffer and separated into two tubes, and incubated on ice for 10 mins followed by subsequent sonication of lysates at 9W, 10 s ON/30 s OFF, nine cycles. Lysates were collected by centrifugation at 10,000 rpm for 10 mins at 4°C. The supernatant (single nucleosome lysate) was transferred to a new tube, and stored at −80°C for the following experiments. 50 μl of lysate was used for the analysis of chromatin digestion.

*Pulldown:* The single nucleosome lysate was diluted with ChIP buffer with protease inhibitor cocktail (for each pulldown, 200 μl lysate was added into 300 μl 1 × ChIP Buffer). 10 μl of sample was saved as input. To reduce nonspecific binding, the lysate was pre-cleared with 10% BSA and 30 μl ChIP Grade Protein G Magnetic Beads (CST, #9006) at 4°C with rotation for 1 hr. The supernatant was collected by placing the tube on a magnetic separation rack. Antibodies of interest or IgG control were added and incubated on rotation at 4°C overnight followed by incubation with 30 μl of magnetic beads for 2 hr at 4°C. Beads were washed with low salt wash buffer (100 μl 10 × ChIP Buffer in 900 ml water) three times and with high salt wash buffer (100 μl 10 × ChIP Buffer and 70 μl 5 M NaCl in 830 μl water) once on rotation.

The antibodies used in ChIP pulldown were: anti-FXR1 (Sigma, HPA018246, RRID: AB_1849204); anti-FXR2 (Invitrogen, PA5-28979, RRID: AB_2546455); anti-STAT1 (Santa Cruz, sc-345, RRID: AB_ 675903; CST, 14994S); anti-STAT3 (Santa Cruz, sc-482, RRID: AB_632440; CST, 12640S, RRID: AB_ 2629499); rabbit IgG (CST, 2729, RRID: AB_1031062); anti-H3K4me3 (CST, 9727, RRID: AB_561095); anti-H3K9me3 (Active Motif, 39161, RRID: AB_2532132); and anti-H3K27me3 (Millipore, 07–449, RRID: AB_310624).

For ChIP-WB, protein G beads were added with 30 μl of DTT-containing 2 × SDS sample buffer and denatured at 95°C in thermomixer with gentle vortexing (1,200 rpm) for 5 min. Eluted proteins were separated in SDS-PAGE gel and analyzed by WB.

For ChIP-MS, eluted proteins were separated in SDS-PAGE gel followed by gel fixing in buffer containing methanol, acetic acid, and water at 5:1:4 ratio. The gel was stained in EZBlue Gel Staining Reagent (Sigma, G1041) and subjected to mass spectrometry (MS) analysis.

For ChIP-seq or ChIP-PCR, beads were eluted with 150 µl of ChIP elution buffer for 30 mins at 65°C with gentle vortexing (1,200 rpm) in thermomixer. Eluted supernatant was incubated with 6 µl 5 M NaCl and 2 µl Proteinase K at 65°C for 2 hr in thermomixer to digest the protein. DNA was purified using spin columns, and sent for ChIP-seq or ChIP-PCR analysis.

## ChIP-seq and data analysis

DNA from ChIP pull down complexes was subjected to library preparation according to Illumina's instructions (Part # 11257047). Sequencing was performed on Illumina HiSeq 2000. ChIP-seq data quality was analyzed using FastQC (*Andrews, 2010*). Reads in the fastq files were trimmed by 5 bp in the 5′-end, and then mapped to the reference human genome (HG19) using BOWTIE2. Multiple mapped reads and duplicate reads were removed. FXR1 binding genomic regions (called ChIP-seq peaks) were identified by the Model-based Analysis of ChIP-Seq (MACS) algorithm using the default p-value cutoff of $1 \times e^{-5}$. The distribution of FXR1 ChIP-seq peaks were investigated by the annotations on the human genome. The peaks were binned into eight groups according to the positions relative to annotated genes: 5′ Distal (up to 100 kb upstream TSS), 5′ Proximal (up to 10 kb upstream TSS), 5′ TSS (5 kb upstream TSS to 1 kb downstream TSS), Intragenic (1 kb downstream TSS to 1 kb upstream TTS), 3′ TTS (1 kb upstream TTS to 5 kb downstream TTS), 3′ Proximal (up to 10 kb downstream TTS), 3′ Distal (up to 100 kb downstream TTS), and the remaining sequence defined as Intergenic region. The aggregation signals of FXR1, H3K4me3, STAT1, and STAT3 at genebody (TSS to TTS) plus upstream and downstream 5 kb region were extracted and plotted using in-house perl and MATLAB code. A hub for the UCSC genome browser was constructed in order to investigate the detailed information of the peaks. The summit positions of the FXR1 ChIP-seq peaks were extended to 100 bp both upstream and downstream as input region for HOMER (*Heinz et al., 2010*) to call known and de novo motifs. The overlap between ChIP-seq peaks was determined by 1 bp overlap of two peaks that were extended by 500 bp both upstream and downstream. The TSS associated genes in each ChIP-seq were determined by overlapping peaks to refseq genes coordinates. Venn diagrams were generated using online web tool VENNY 2.1 (http://bioinfogp.cnb.csic.es/tools/venny/index.html). Protein function enrichment was analyzed using DAVID (*Huang et al., 2008, 2009*). The ChIP-seq peaks were visualized using IGV (http://www.broadinstitute.org/igv/).

The sequence of ChIP-PCR primers is listed in *Supplementary file 8*.

## Chromatin immunoprecipitation followed with mass spectrometry (ChIP-MS) and data analysis

FXR1 or IgG control ChIP pulldown complexes were separated in SDS-PAGE gel. The gel was stained with EZblue (Sigma, G1041). The stained gel was cut into multiple parts based on protein molecular weight. Gel pieces were destained in solution containing 50 mM $NH_4HCO_3$ (Sigma Aldrich) and Acetonitrile (ACN) (Fisher Scientific) at 1:1 ratio, dried with SpeedVac, rehydrated in 300 µl of 10 mM DTT (Sigma Aldrich) solution at 56°C for 1 hr, and reacted in 25 mM IAA (Sigma Aldrich) at 37°C for 30 mins. After washing with 25 mM $NH_4HCO_3$ (Sigma Aldrich) for 10 mins, the gel pieces were digested with trypsin solution (0.01 g/L trypsin in 25 mM $NH_4HCO_3$) (Promega) at 37°C for 12 hr. The digested peptides were extracted using 300 µl of 50% ACN-0.1% FA. The extracted solution was collected and concentrated with SpeedVac. The samples were resolved with 0.1% FA and analyzed by nanoLC-MS/MS (Thermofisher). MS/MS was performed in a data-dependent mode in which the top 10 most abundant ions (S/N > 30) for each MS scan were selected for MS/MS analysis by HCD. Parameters used during searching included enzyme, trypsin; missed cleavages, One; dynamic modifications, oxidation (M); static modifications, Carbamidomethyl (C); precursor mass tolerance, 10 ppm; fragment mass tolerance, 0.1 Da. Parameters in data filtration with Prohits included kinds of exclusion proteins: Heat Shock, Ribosomal, Cytoskeleton, Keratin, Artifact Protein, Rib, Nucleoprotein and Albumin; Max SAINT < 0.7 (*Liu et al., 2012*; *Liu et al., 2010*). All MS and MS/MS data were searched in the combined mode against Uniprot Human database (August, 2013) using the Proteome Discoverer 1.3 software (Thermofisher). A score greater than 31

(confidence interval values > 95%) obtained with the Mascot search engine (Matrix Science, U.K.) was considered as significant. All the searching results were filtered with Prohits software to remove non-specific binding proteins (Max SAINT < 0.7) ( *Liu et al., 2010*,  *2012*).

FXR1 interacting proteins were classified according to the GO (gene ontology) database. Protein function clustering was analyzed with DAVID. Visualization and network map of FXR1-interacting proteins were delineated using BioGRID in Cytoscape. Protein–protein interaction was analyzed using the KEGG (Kyoto Encyclopedia of Genes and Genomes) pathway database.

## RNA extraction and quantitative RT-PCR (q-RT-PCR)

Total RNA was extracted using RNeasy Mini Kit (Qiagen, 74106) and Qiagen QIAcube 230 V w. starter Pack-2 machine, and reverse-transcribed using Oligo (dT) primer and the SuperScript III First-Strand synthesis system (Invitrogen, 18080–051). q-RT-PCR using cDNA product as the template was performed using Power SYBR Green PCR Master Mix (Applied Biosystems (AB), 4367659) and an ABI 7900HT Fast Real-Time PCR-3 Machine with specific primers. Gene expression was quantified using the comparative Ct (cycle threshold) method and the results were normalized to GAPDH. Sequences of q-RT-PCR primers are listed in *Supplementary file 8*.

## RNA-seq

Each RNA-seq sample had three biological replicates. RNA isolation was performed using Trizol and RNeasy MinElute Cleanup Kit from cells. Following extraction, RNA quantity was assessed using Nanodrop and the integrity of total RNA using the RNA 6000 Nano Kit on an Aligent 2100 Bioanalyzer. An RNA library was constructed using the TruSeq RNA Sample Preparation Kit. The first step in the workflow involved purifying the poly-A containing mRNA molecules using oligo-dT-attached magnetic beads. Following purification, mRNA was fragmented into small pieces using divalent cations under elevated temperature. The cleaved RNA fragments were copied into first-strand cDNA using reverse transcriptase and random primers. Second-strand cDNA synthesis followed, using DNA PolymeraseI and RNase H. The cDNA fragments then went through an end repair process, involving the addition of a single 'A' base, and then ligation of the adapters. The products were then purified and enriched with PCR to create the final cDNA library. The clusters of the cDNA library were generated on HiSeq X HD PE flow cell. Then the clusters were sequenced on a HiSeq X system. RNA-seq data quality was analyzed using FastQC (*Andrews, 2010*). Reads in the fastq files were trimmed by five base pairs at the 5′-end, and then mapped to the reference human genome (HG19) with the algorithm STAR using default parameters. Multiple mapped reads and duplicate reads were removed to extract raw reads of refseq genes using HOMER. Differential expressed genes were identified using R package DESeq2 and determined using $log_2FC$ cutoff 0.6 and adjusted p value cutoff 0.01. KEGG pathway enrichment was investigated using GSEA (*Subramanian et al., 2005*; *Mootha et al., 2003*).

## Histone methylated lysine analog (MLA) protein pulldown assay

Histone H3 MLA proteins with methylation at various residues were purchased from Active Motif (31208–31222). His-tagged-FXR1-Tudor (2-132) protein were expressed and purified from an *Escherichia coli* system, and pulldown assay was performed as previously described (*Alpatov et al., 2014*). In brief, in each reaction, 2 μg of MLA protein and 2 μg of FXR1-Tudor protein were added in 500 μl of Histone MLA Binding Buffer (50 mM Tris pH 7.5, 1 M NaCl, 2 mM $MgCl_2$, 0.5% Triton X-100, and 10 mM Imidazole) (save 10 μl as input), followed by rotation at 4℃ for 30 mins. The reaction solution was spun down (10,000 g, 2 min) to collect the supernatant in a new tube. 15 μl of Ni-NTA beads (Qiagen) was added to the collected supernatant followed by rotation at 4℃ for 1 hr. Beads were collected by spinning at 1,000 g for 2 min, washed with 1 ml of Histone MLA Binding Buffer five times, suspended in 30 μl of 2 × SDS sample buffer and boiled to denature the proteins. The FXR1-bound histone MLA proteins were detected in a WB assay.

## JAK inhibitor assay

JAK inhibitor S-Ruxolitinib (dual JAK1/2 inhibitor tool compound) was purchased from selleck (http://www.selleck.cn/). Owing to the instability of the compound, the medium with inhibitor was replaced every 5 days.

## siRNA-mediated gene knockdown

Gene-specific siRNA sequence was designed using the software in Hannon Lab (http://cancan.cshl.edu/RNAi_central/RNAi.cgi?type=siRNA). The siRNA oligonucleotides were produced by Invitrogen, and the MISSION siRNA Universal Negative Control #2 (Sigma) was used as a negative control. Five siRNAs for each gene were tested. siRNAs were introduced into cells at a final concentration of 5–20 nM using Lipofectamine RNAiMAX Transfection Reagent (Invitrogen, 13778–150) and Opti-MEM (Gibco, 31985–062) according to the manufacturer's protocol. Knockdown efficiency was assessed 48 hr post-transfection by q-RT-PCR or WB. Sequences of siRNAs are listed in *Supplementary file 8*.

## In vitro pull down assay for FXR1-STAT interaction

Flag-tagged FXR1 or HA-tagged-STAT1 or -STAT3 proteins were expressed in HEK 293 T cells. Cell lysates were prepared using lysis buffer (50 mM Tris-Cl, pH 7.5, 500 mM NaCl, 1% Triton X-100) with complete protease inhibitor cocktail (Sigma, P8340), phosphatase inhibitor cocktail (Sigma, P5726), and PMSF (Sigma, 93482–50 ml-F). Lysate was incubated, at 4°C overnight, with 100 µl Flag-M2 magnetic beads (Sigma, M8823) to purify Flag-tagged-FXR1 protein or with 100 µl Influenza Hemagglutinin (HA) magnetic beads (Thermo Fisher, 88837) to purify HA-tagged-STAT protein. Beads were washed with 1 ml lysis buffer three times and with 600 µl elution buffer (50 mM Tris-Cl, pH7.5, 500 mM NaCl, 10% Glycerol) once. Target protein was eluted with 30 µl of Flag peptide solution (5 mg/ml, in elution buffer, Sigma, F4799-4MG) to obtain Flag-tagged-FXR1 protein or with HA peptide (2 mg/ml, in elution buffer, Sigma, I2149-1MG) to harvest HA-tagged-STAT protein.

For in vitro pulldown, 25 µl purified Flag-tagged-FXR1 protein was incubated with 75 µl purified HA-tagged-STAT1 or -STAT3 protein in 500 µl binding buffer (50 mM Tris pH 7.5, 150 mM NaCl, 0.1% NP-40, 1 mM DTT, phosphatase inhibitor and protease inhibitor cocktail) at 4°C overnight. 30 µl Flag or HA magnetic beads were added and incubated at 4°C for another 3 hr. Beads were washed with wash buffer (50 mM Tris-pH 7.5, 300 mM NaCl, 0.1% NP-40) three times. The eluted products in 30 µl elution buffer were subjected to WB analysis.

## mRNA binding assay

H358 cells were lysed in lysis buffer (100 mM NaCl, 10 mM $MgCl_2$, 30 mM Tris-HCl, pH 7.5, 1 mM dithiothreitol [DTT], protease inhibitor cocktail, 100 U/ml RNasin, 0.1% Nonidet P40 [NP40]) and divided into two parts. 2% of lysate was saved as input. The lysate was pre-cleared by 10% BSA and 30 µl ChIP Grade Protein G Magnetic Beads (CST, #9006) on rotation for 1 hr at 4°C. The supernatant was collected by a magnetic separation rack, incubated with FXR1 antibody (Sigma, HPA018246) or IgG control (CST, 2729) for 1 hr at 4°C, and incubated with 30 µl of magnetic beads for 1 hr at 4°C on rotation. Beads were washed with NT2 buffer (50 mM Tris-Cl, 150 mM NaCl, 1 mM $MgCl_2$, and 0.05% NP-40) five times. Endogenous RNA protein complexes (RNP) were eluted from antibody/Protein G beads by incubation with 150 µl of ChIP elution buffer for 30 mins at 65°C with gentle vortexing (1200 rpm) in thermomixer. Proteins in the elution were digested by Proteinase K. RNA was purified using spin columns from RNeasy Mini Kit (Qiagen, 74106) and reverse-transcribed using Oligo (dT) primer and the SuperScript III First-Strand synthesis system (Invitrogen, 18080–051). The cDNA was subjected to RT-PCR assay using AccuPrime Pfx SuperMix (Invitrogen, 12344040) (*Qian et al., 2015*). The PCR products are detected by electrophoresis in agarose gels and stained with ethidium bromide, visualized by ChemiDoc. The non-target genes *GAPDH and Actin* serve as the negative controls.

## Accession number

ChIP-seq data reported in this paper have been deposited in the Gene Expression Omnibus (GEO) database under accession number GSE79707. RNA-seq data reported in this paper have been deposited in the Gene Expression Omnibus (GEO) database under accession number GSE101754.

## Acknowledgements

We thank the Postdoc Committee at China Novartis Institutes for BioMedical Research for helpful discussion and advice. We thank Witold Filipowicz (FMI) for critical reading of the manuscript and for

valuable suggestions. We thank Elvin Wang, Hejun Liu, Liqing Geng, and Ming Fang for technical support.

## Additional information

### Competing interests

Jiao Yue: Jiao Yue is an employee for Novartis, Inc., where part of the study was conducted. Mengtao Xiao: Mengtao Xiao is an employee for Novartis, Inc., where part of the study was conducted. Han Han-Zhang: Han Han-Zhang is an employee for Novartis, Inc., where part of the study was conducted. Yanyan Yu: Yanyan Yu is an employee for Novartis, Inc., where part of the study was conducted. Shen Niu: Shen Niu is an employee for Novartis, Inc., where part of the study was conducted. Youjia Hua: Youjia Hua is an employee for Novartis, Inc., where part of the study was conducted. Peter Atadja: Peter Atadja is an employee for Novartis, Inc., where part of the study was conducted. The other authors declare that no competing interests exist.

### Funding

| Funder | Grant reference number | Author |
|--------|------------------------|--------|
| Novartis | Research | Bin Xiang |
| Novartis | Postdoc program | Yichao Fan |

The authors declare that there was no funding for this work.

### Author contributions

Yichao Fan, Conceptualization, Resources, Data curation, Formal analysis, Supervision, Validation, Investigation, Methodology, Writing—original draft, Project administration, Writing—review and editing; Jiao Yue, Resources, Formal analysis, Validation, Investigation, Methodology; Mengtao Xiao, Resources, Validation, Investigation, Methodology; Han Han-Zhang, Validation, Writing—review and editing; Yao Vickie Wang, Data curation, Formal analysis, Visualization; Chun Ma, Resources, Data curation, Investigation, Methodology; Zhilin Deng, Data curation, Investigation; Yingxiang Li, Zhiping Weng, Formal analysis, Visualization; Yanyan Yu, Visualization, Methodology; Xinghao Wang, Methodology; Shen Niu, Software; Youjia Hua, Software, Formal analysis; Peter Atadja, Conceptualization, Project administration; En Li, Conceptualization, Resources, Data curation, Supervision, Project administration, Writing—review and editing; Bin Xiang, Conceptualization, Resources, Data curation, Formal analysis, Supervision, Funding acquisition, Validation, Investigation, Methodology, Writing—original draft, Project administration, Writing—review and editing

### Author ORCIDs

Yingxiang Li, http://orcid.org/0000-0003-0835-9280
Bin Xiang, http://orcid.org/0000-0002-6973-100X

### Ethics

Animal experimentation: All the procedures related to animal handling, care and the treatment in the study were performed according to the guidelines approved by the Institutional Animal Care and Use Committee (IACUC) of WuXi AppTec following the guidance of the Association for Assessment and Accreditation of Laboratory Animal Care (AAALAC). The approved protocol number is R20150728-Mouse and Rat. The animals were daily checked for any effects of tumor growth and treatments on normal behavior such as mobility, food and water consumption, body weight gain/loss, eye/hair matting and any other abnormal effects. Death and observed clinical signs were recorded.

### Decision letter and Author response

Decision letter https://doi.org/10.7554/eLife.26129.055
Author response https://doi.org/10.7554/eLife.26129.056

# Additional files

## Supplementary files

• Supplementary file 1. FXR1 potential interacting proteins predicted by ChIP-MS in KATOIII and H358 cell lines.
DOI: https://doi.org/10.7554/eLife.26129.047

• Supplementary file 2. Function clustering of the FXR1 potential interacting proteins using the GO and DAVID analysis.
DOI: https://doi.org/10.7554/eLife.26129.048

• Supplementary file 3. FXR1, FXR2, histone marks and STATs ChIP-seq peaks, distribution, and overlap analysis.
DOI: https://doi.org/10.7554/eLife.26129.049

• Supplementary file 4. Table S4-GO pathway analysis of FXR1-H3K4me3 or FXR1-STATs overlapped or non-overlapped ChIP-seq target genes in H358 cells.
DOI: https://doi.org/10.7554/eLife.26129.050

• Supplementary file 5. Target gene validation-RT-PCR-primers.
DOI: https://doi.org/10.7554/eLife.26129.051

• Supplementary file 6. FXR1 target gene analysis using RNA-seq in H358 cells.
DOI: https://doi.org/10.7554/eLife.26129.052

• Supplementary file 7. Gene expression profile of genes with FXR1 occupancy at promoter.
DOI: https://doi.org/10.7554/eLife.26129.053

• Supplementary file 8. Reagent information.
DOI: https://doi.org/10.7554/eLife.26129.054

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
