## [Decision Letter]

Thank you for submitting your article "FXR1 regulates transcription and is required for tumor growth in TP53 homozygous deletion cancers" for consideration by *eLife*. Your article has been favorably evaluated by Kevin Struhl (Senior Editor) and three reviewers, one of whom, Irwin Davidson, is a member of our Board of Reviewing Editors. The following individual involved in review of your submission has agreed to reveal their identity: Pierre Massion (Reviewer #2).

The reviewers have discussed the reviews with one another and the Reviewing Editor has drafted this decision to help you prepare a revised submission.

Summary:

This study reports that FXR1 plays an essential role in the proliferation of tumour cells bearing a homozygous double TRP53-FXR2 deletion. The paper shows that knockdown of FXR1 in cells with the double deletion arrests their proliferation. The authors go onto show that FRX1 binds to a small number of promoters and facilitates the recruitment of STAT1 that activates their expression and that some of these target genes are essential for the proliferation of the TRP53-FXR2 deleted cells. The reviewers found that the data are novel and move the field forward, strengthening the evidence for targeting FXR1 in cancers where TRP53 and FXR2 are deleted.

Essential revisions:

The additional experiments that should be provided concern two main points. The relationship between FXR1 and FXR2 function and secondly the crosstalk between FXR1 and STAT1/3 that is not properly established. The reviewers are aware that these may take more than two months to complete, but they will be required.

Cells with a double TRP53/FXR2 double deletion are dependent on FXR1. The authors should use CRISPR/Cas9 to make a TRP53/FXR1 double deletion and show that the resulting cells are now FXR2 dependent. This would strengthen the idea that FXR1 and FXR2 fulfill at least partially redundant and essential functions. For example, the authors show that depletion of FXR2 reduces expression of the FXR1 target gene CASC4 in cells where FXR1 is normally expressed (Figure 5). This result rather suggests that FXR1 cannot compensate for FXR2 depletion at this gene and that there would appear to be no redundancy between FXR1 and FXR2 at least for expression of this gene.

Along the same lines: the authors performed FXR1 ChIP-seq and they performed FXR2 ChIP-qPCR. As they have a ChIP-grade FXR2 antibody, they should perform FXR2 ChIP-seq to determine the overlap in genomic binding and function of FXR1 and FXR2.

According to the ChIP-seq experiments FXR1 has only a limited number of binding sites around half of which were also marked by H3K4me3. What about the sites that showed no H3K4Me3, what do these sites correspond to, promoters of repressed genes? Also, as only 37-49% of FXR1 sites correspond to STAT-binding sites, what do the others correspond to? The authors should perform motif enrichment analyses (like for example MEME ChIP) to identify other transcription factors that may be enriched at these sites. Also without performing STAT1/3 ChIP-seq how can the authors be sure that there is no STAT protein present at these sites in these cells? More importantly, the authors state that only a small subset of STAT sites is co-occupied by FXR1. The relationship between FXR1 and the STATs is therefore not clear and the factors that dictate FXR1 genomic binding remain unknown. There are perhaps other factors that recruit FXR1, that may or may not have been detected by mass-spectrometry. A more thorough bioinformatic analyses of the FXR1 binding sites is required. Otherwise one could argue that FXR1 acts as a cofactor for a transcription factor, other than STAT1/3, present at almost all target genes.

In relation to this, there is some inconsistency between the results shown in Figure 5—figure supplement 6 and Figure 6—figure supplement 1/B. Treatment of H358 cells with the JAK1 inhibitor at 1 µM completely blocks GLI1 expression (Figure 5) and knocking down GLI1 partly inhibits cell growth. (Figure 6—figure supplement 1). However, incubating the cells with the JAK1 inhibitor does not affect cell proliferation. Does this mean that STAT1/3 activity is not needed for the proliferation of these cells? Also, does suppression of STAT1/3 affect FXR1 expression?

Figure 5, Figure 5—figure supplement 5: The authors state: 'some FXR1 target genes are inhibited by the JAK1 inhibitor'. Are these the genes in which both STAT1/3 and FXR1 can be found on the promoter? Is, for example, FXR1 binding to the SSBP2 promoter but not STAT1/3? Or why is expression of SSBP2 not down-regulated by S-Ruxolitinib?

Some of these issues could be clarified by performing RNA-seq to get an idea of the overall effects of FXR1 silencing on gene expression. Are there a small number of affected genes perhaps in line with the small number of occupied sites? Or alternatively, FXR1 silencing may have widespread effects on gene expression that go well beyond the small number of affected genes investigated. If so how could this be explained?

According to IARC database (http://p53.iarc.fr/CellLines.aspx), and the literature (https://www.ncbi.nlm.nih.gov/pubmed/2155427), Hep3B should be a p53-null cell line. Can the authors comment on this and have they verified the identity of the Hep3B cells they used?

---

## [Author Response]

Essential revisions:The additional experiments that should be provided concern two main points. The relationship between FXR1 and FXR2 function and secondly the crosstalk between FXR1 and STAT1/3 that is not properly established. The reviewers are aware that these may take more than two months to complete, but they will be required.Cells with a double TRP53/FXR2 double deletion are dependent on FXR1. The authors should use CRISPR/Cas9 to make a TRP53/FXR1 double deletion and show that the resulting cells are now FXR2 dependent. This would strengthen the idea that FXR1 and FXR2 fulfill at least partially redundant and essential functions. For example, the authors show that depletion of FXR2 reduces expression of the FXR1 target gene CASC4 in cells where FXR1 is normally expressed (Figure 5). This result rather suggests that FXR1 cannot compensate for FXR2 depletion at this gene and that there would appear to be no redundancy between FXR1 and FXR2 at least for expression of this gene.

We thank the reviewer for this valuable suggestion. To further strengthen the redundant function shared between FXR1 and FXR2, we generated AGS *TP53/FXR1* double knockout (DKO) cells using CRISPR-Cas9, and stably expressed FXR2 shRNA-2 or FXR2 shRNA-3 in these DKO cells. We observed that knockdown of FXR2 upon Dox induction suppressed proliferation in these cells (added in Figure 2—figure supplement 3), confirming the redundant function shared between FXR1 and FXR2. Such redundancy was also observed in previous study of FXR1 knockdown-induced anti-proliferation effect in *TP53/FXR2* DKO cells.

As pointed out by the reviewer in Figure 5, FXR2 knockdown (KD) partially inhibited the expression of target gene *CASC4* in AGS cells where FXR1 was normally expressed. We further evaluated the effect of FXR2 KD on the expression of target genes in AGS cells. We observed that FXR2 KD in general had limited impact on the expression of the target genes except for slight inhibition of *CASC4* and *TRAPPC9* (revised in Figure 5). This suggested that FXR1 in AGS cells compensated for FXR2 KD at least partially on regulating the target genes transcription. We cannot rule out the possibility that FXR1 and FXR2 don’t have complete redundant function in regulating certain genes for example CASC4 in AGS cells.

Along the same lines: the authors performed FXR1 ChIP-seq and they performed FXR2 ChIP-qPCR. As they have a ChIP-grade FXR2 antibody, they should perform FXR2 ChIP-seq to determine the overlap in genomic binding and function of FXR1 and FXR2.

We conducted FXR1, FXR2, STAT1, STAT3, and H3K4me3 ChIP-seq in AGS cells. The data indicated that the four proteins were enriched at TSS region in the genome (added in Figure 5, Figure 5—figure supplement 4 and Supplementary file 3). 89.95% (2935/3263 FXR1 peak-associated genes) FXR1 peak-associated genes overlap with FXR2-occupied genes in AGS cells (added in Figure 5), and the percentage increased when we focused at TSS (Figure 5—figure supplement 4) suggesting a shared function of FXR1 and FXR2 at gene promoters. However, only a small percentage of FXR2 binding site or peak genes overlap with FXR1’s in AGS cells. The gap is likely due to the weaker capability of FXR1 ChIP antibody in pull down assay comparing to FXR2 ChIP antibody which is reflected by the difference in in peak-associated gene numbers of FXR1 and FXR2 (3,263 and 15,346), respectively in AGS ChIP-seq study (added in Figure 5). However, this doesn’t change the conclusion that FXR1 binding sites largely overlap with FXR2 sites.

Furthermore, we examined FXR1 and FXR2 co-localization at promoters of target genes by ChIP-PCR (added in Figure 5 and Figure 5—figure supplement 4). We observed overlapped binding of FXR1, FXR2, STAT1, and STAT3 at promoters of seven target genes in AGS cells (added in Figure 5 and Figure 5—figure supplement 4).

Collectively, evidences supporting redundant function shared between FXR1 and FXR2 in this study include: 1) both FXR1 and FXR2 can rescue FXR1 shRNA-inhibited cell proliferation (Figure 1), 2) They share a significant portion of overlapping binding sites in the genome (Figure 5, Figure 5—figure supplement 4, and Supplementary file 3), 3) Knockout of either one sensitizes TP53 homozygous deletion-containing cancer cells for the inhibition of the remaining member(Figure 2, added in Figure 2—figure supplement 3). Their redundancy is also supported by previous reports from other labs that FXR1 and FXR2 have common function in mRNA binding.

According to the ChIP-seq experiments FXR1 has only a limited number of binding sites around half of which were also marked by H3K4me3. What about the sites that showed no H3K4Me3, what do these sites correspond to, promoters of repressed genes? Also, as only 37-49% of FXR1 sites correspond to STAT-binding sites, what do the others correspond to? The authors should perform motif enrichment analyses (like for example MEME ChIP) to identify other transcription factors that may be enriched at these sites. Also without performing STAT1/3 ChIP-seq how can the authors be sure that there is no STAT protein present at these sites in these cells? More importantly, the authors state that only a small subset of STAT sites is co-occupied by FXR1. The relationship between FXR1 and the STATs is therefore not clear and the factors that dictate FXR1 genomic binding remain unknown. There are perhaps other factors that recruit FXR1, that may or may not have been detected by mass-spectrometry. A more thorough bioinformatic analyses of the FXR1 binding sites is required. Otherwise one could argue that FXR1 acts as a cofactor for a transcription factor, other than STAT1/3, present at almost all target genes.

We thank the reviewer for raising this point that only about half of FXR1 binding sites overlapped with H3K4me3. We found that the majority of the overlapped peaks located at TSS region (added in Figure 5, Figure 5—figure supplement 1 and Supplementary file 3). We analyzed FXR1 sites with no H3K4me3 localization and revealed that these sites more correspond to 5’ distal, intergenic, and 3’ distal region (added in Figure 5—figure supplement 1 and Supplementary file 3). From the Go analysis results, we found that while the FXR1-H3K4me3 overlapped genes function in multiple pathways, there is no significant pathway clustering result for the non-overlapped genes (Supplementary file 4).

Upon the reviewer’s comment on our conclusion regarding FXR1 and STAT relationship, we conducted STAT1 and STAT3 ChIP-seq in the same H358 cells to re-evaluate FXR1 and STAT genomic binding instead of using STAT ChIP-seq database as in the original manuscript. Our ChIP-seq results are in an agreement with our previous results obtained from Encode database. STAT1 and STAT3 showed sporadic binding in the genome of H358 cells. In the same cells, about 24.38% (215 in 882) and 25.74% (227 in 882) of FXR1 binding sites overlap with STAT1 and STAT3 binding, respectively (Figure 5—figure supplement 1, and Supplementary file 3). Especially at TSS, 64.65-65.66% of FXR1 peaks overlap with STAT binding suggesting that FXR1 mainly engage in STAT1/3 in transcription (Figure 5—figure supplement 1, and Supplementary file 3). Overall, a total of 493 peak-associated genes were identified for FXR1 (Figure 5 and Supplementary file 3). Among them, 319 and 373 peak-associated genes overlapped with STAT1 and STAT3 respectively (Figure 5 and Supplementary file 3). Overall, a total of 493 FXR1 peak-associated genes were identified (Figure 5 and Supplementary file 3). Among them, 319 and 373 peak-associated genes overlapped with STAT1 and STAT3, respectively (Figure 5 and Supplementary file 3). At TSS, a total of 210 peak-associated genes were identified for FXR1 (Figure 5 and Supplementary file 3). Among them, 145, 147 and 200 peak-associated genes overlapped with STAT1, STAT3 and H3K4me3, respectively (Figure 5 and Supplementary file 3). More analyses of FXR1 and STAT relationship in H358 cells were presented in Figure 5—figure supplement 1.

Further, the majority of the FXR1-STATs overlapped peaks located at TSS region (Figure 5, Figure 5—figure supplement 1 and Supplementary file 3). We also found that FXR1 sites with no STAT binding more correspond to 5’ distal, intragenic, and 3’ distal region (Figure 5—figure supplement 1 and Supplementary file 3). Motif analysis identified STAT-binding motif in FXR1- and STAT-shared binding sites (added in Figure 5—figure supplement 1). And, the no STAT binding sites contain GATAs and other binding motif (added in Figure 5—figure supplement 1).

ChIP-MS and follow-up assays revealed a novel function of FXR1, mediating transcription by interacting with transcription factor STAT1/3. We conducted more studies to determine FXR1’s role in regulating STAT binding at gene promoters in H358 cells. Consistent with the observation in the original manuscript, knockdown of FXR1 inhibited the binding of FXR1, STAT1, and STAT3 at the promoter regions of all seven target genes in ChIP-PCR study (revised in Figure 5). Our data suggested that FXR1 participates in STAT1/3 transcription likely by recruiting or stabilizing STAT to the promoter of genes. In regions where co-localization of FXR1 and STAT was not observed, FXR1 may interact with other transcription factor(s) such as GATAs indicated from the binding motif analysis (added in Figure 5—figure supplement 1). We cannot rule out the possibility that our ChIP-seq didn’t capture all binding sites of FXR1 and STAT due to the assay limitations. In our ChIP-MS study, another transcription factor BTF (BCl^-^2-associated transcription factor 1) was revealed in FXR1 protein complex (Supplementary file 1). Further investigation is needed to determine whether FXR1 interacts with BTF to regulate transcription at gene promoters. It is also possible that our ChIP-MS study missed other transcription factor(s) that interacts with FXR1 at non STAT binding sites in regulating transcription.

In relation to this, there is some inconsistency between the results shown in Figure 5—figure supplement 6 and Figure 6—figure supplement 1/B. Treatment of H358 cells with the JAK1 inhibitor at 1 µM completely blocks GLI1 expression (Figure 5) and knocking down GLI1 partly inhibits cell growth. (Figure 6—figure supplement 1). However, incubating the cells with the JAK1 inhibitor does not affect cell proliferation. Does this mean that STAT1/3 activity is not needed for the proliferation of these cells? Also, does suppression of STAT1/3 affect FXR1 expression?

Our ChIP-seq results revealed that STAT1/3 have 15639 or 28739 binding sites cross the genome in H358 cells, suggesting that STATs regulate the expression of a spectrum of genes. Therefore, the inhibitor of its kinase JAK could affect the expression of many genes. *GLI1* is one of FXR1 target genes whose transcription is regulated by FXR1 and STAT. Given FXR1 only involves in a small portion of STAT binding sites, *GLI1* could be one of many JAK-STAT target genes. GLI1 knockdown inhibited cell proliferation suggesting that the cells are dependent on FXR1-GLI1 signaling. JAK inhibitor didn’t block cell proliferation in the same cells suggesting that the cells are not dependent on JAK-STAT signaling for cell growth. While GLI1 expression was inhibited, JAK inhibitor could impact on many genes whose function in regulating cell growth in the tested cells are not investigated. The different effects of JAK inhibitor or GLI1 siRNA on cell proliferation do not necessarily suggest inconsistency, rather it indicates the complexity of JAK-STAT signaling in regulating cell growth. We didn’t observe changes in FXR1 expression upon JAK inhibitor treatment (added in Figure 5).

Figure 5, Figure 5—figure supplement 5: The authors state: 'some FXR1 target genes are inhibited by the JAK1 inhibitor'. Are these the genes in which both STAT1/3 and FXR1 can be found on the promoter? Is, for example, FXR1 binding to the SSBP2 promoter but not STAT1/3? Or why is expression of SSBP2 not down-regulated by S-Ruxolitinib?

We conducted ChIP-PCR and observed the co-localization of FXR1 and STAT1/3 at the promoter region of all 7 target genes in both H358 (added in Figure 5) and AGS cells (revised in Figure 5—figure supplement 4). Further investigation is needed to address why JAKi S-Ruxolitinib didn’t inhibit SSBP2 expression. One possibility is that additional regulators are involved in FXR1-STAT transcriptional activity at SSBP2 gene.

Some of these issues could be clarified by performing RNA-seq to get an idea of the overall effects of FXR1 silencing on gene expression. Are there a small number of affected genes perhaps in line with the small number of occupied sites? Or alternatively, FXR1 silencing may have widespread effects on gene expression that go well beyond the small number of affected genes investigated. If so how could this be explained?

We performed RNA-seq to analyze gene expression in H358 cells with biological triplicates (added in Supplementary file 6). We selected FXR1 knockdown-induced gene expression changes using log2 fold change >0.6 as cutoff (Supplementary file 6). Among the 7 FXR1 target genes identified by ChIP-seq, 5 showed significant downregulation upon FXR1 knockdown in H358 cells (added in Supplementary file 7). The remaining two genes (*SSBP2* and *NPAS3*) showed moderate downregulation (Supplementary file 7). Among the genes whose promoters are occupied by FXR1, 45/210 genes showed downregulation upon FXR1 knockdown in RNA-seq assay (added in Figure 5—figure supplement 1 and Supplementary file 7). RNA-Seq revealed significantly more genes regulated by FXR1 than the number of genes revealed by ChIP-seq. The discrepancy may primarily attribute to the fact that FXR1 is a well-known mRNA-binding protein regulating RNA metabolism and stability in addition to the newly discovered function in transcription regulation in this study. Therefore, FXR1 knockdown could impact on both populations of genes that regulated by FXR1’s transcriptional activity and RNA-binding function. In addition, the scope and limitation of RNA-seq and ChIP-seq could also contribute to the discrepancy.

According to IARC database (http://p53.iarc.fr/CellLines.aspx), and the literature (https://www.ncbi.nlm.nih.gov/pubmed/2155427), Hep3B should be a p53-null cell line. Can the authors comment on this and have they verified the identity of the Hep3B cells they used?

Based on the CCLE database, Hep3B carries 1.3 copies of TP53 suggesting a partial deletion. Our WB and Q-PCR assay also indicated the expression of p53 mRNA and protein in Hep3B, while FXR2 level is undetectable. Therefore, Hep3B served as a cell model of FXR2 single deletion in this study (Figure 1—figure supplement 2).